# ORMDL3 restrains type I interferon signaling and anti-tumor immunity by promoting RIG-I degradation

Qi Zeng[1], Chen Yao[1], Shimeng Zhang[1], Yizhi Mao[1], Jing Wang[1], Ziyang Wang[2], Chunjie Sheng[1]\*, Shuai Chen[1]\*

[1]State Key Laboratory of Oncology in South China, Guangdong Provincial Clinical Research Center for Cancer, Sun Yat-sen University Cancer Center, Guangzhou, China; [2]Center for Translational Medicine, The First Affiliated Hospital, Sun Yat-sen University, Guangzhou, China

## eLife Assessment

This study provides **important** insights into the regulation of type-I interferon signaling and anti-tumor immunity, demonstrating that ORMDL3 promotes RIG-I degradation to suppress immune responses. The evidence is **convincing**, with well-executed mechanistic experiments and in vivo validation in syngeneic tumor models. These findings have significant implications for cancer immunotherapy, highlighting ORMDL3 as a potential therapeutic target.

\*For correspondence:
shengchj@sysucc.org.cn (CS);
chenshuai@sysucc.org.cn (SC)

**Competing interest:** The authors declare that no competing interests exist.

**Abstract** Mounting evidence has demonstrated the genetic association of ORMDL sphingolipid biosynthesis regulator 3 (ORMDL3) gene polymorphisms with bronchial asthma and a diverse set of inflammatory disorders. However, its role in type I interferon (type I IFN) signaling remains poorly defined. Herein, we report that ORMDL3 is a negative modulator of the type I IFN signaling by interacting with mitochondrial antiviral signaling protein (MAVS) and subsequently promoting the proteasome-mediated degradation of retinoic acid-inducible gene I (RIG-I). Immunoprecipitation coupled with mass spectrometry (IP-MS) assays uncovered that ORMDL3 binds to ubiquitin-specific protease 10 (USP10), which forms a complex with and stabilizes RIG-I through decreasing its K48-linked ubiquitination. ORMDL3 thus disrupts the interaction between USP10 and RIG-I, thereby promoting RIG-I degradation. Additionally, subcutaneous syngeneic tumor models in C57BL/6 mice revealed that inhibition of ORMDL3 enhances anti-tumor efficacy by augmenting the proportion of cytotoxic CD8 positive T cells and IFN production in the tumor microenvironment (TME). Collectively, our findings reveal the pivotal roles of ORMDL3 in maintaining antiviral innate immune responses and anti-tumor immunity.

## Introduction

Type I interferon (type I IFN) play a key role in the innate immune response to viral infections. Under viral stimulations, cells produce and release interferons, which induce the transcription of interferon-stimulated genes (ISGs) (*Dalskov et al., 2023*; *McNab et al., 2015*). Besides their critical role in antiviral immune responses, growing evidence suggests that type I IFN produced by malignant tumor cells or infiltrating immune cells also influence the effectiveness of cancer immunotherapy (*Zitvogel et al., 2015*; *Holicek et al., 2024*; *Gajewski et al., 2012*). Many traditional chemotherapeutic drugs, targeted anti-tumor drugs, immunoadjuvants, and oncolytic viruses require intact type I IFN signaling to fully exert their effects (*Yang et al., 2014*; *Zitvogel et al., 2013*). Furthermore, studies have shown

that high intratumoral expression levels of type I IFN or ISGs are associated with positive disease outcomes in cancer patients (*Spaapen et al., 2014*).

In response to viral RNA molecules, RIG-I-like receptors (RLRs) exposed their caspase activation and recruitment domains (CARD) and then cooperate with mitochondrial antiviral signaling protein (MAVS) and TANK-binding kinase 1 (TBK1) to promote the production of type I IFN (*Rehwinkel and Gack, 2020*). TBK1 phosphorylates and activates IRF3 and IRF7, which then induce the expression of type I IFN and various ISGs (*Goubau et al., 2013*; *Hartmann, 2017*). Thus, targeting RLRs can provoke anti-infection activities. Moreover, RLRs also play an important role in anti-tumor immunity. For example, targeting RLRs sensitizes 'immune cold' tumors to become 'immune hot' tumors (*Cheon et al., 2014*). DNA methyltransferase inhibitors upregulate endogenous retroviruses in tumor cells to induce the activation of the RLR-mediated RNA recognition pathway that potentiates immune checkpoint therapy (*Chiappinelli et al., 2015*; *Roulois et al., 2015*). SB9200 (also known as inarigivir soproxil or GS9992), an orally available prodrug of a dinucleotide agonist of RIG-I and nucleotide binding oligomerization domain-containing protein 2 (NOD2) (*Korolowicz et al., 2016*), is currently in clinical trials to treat chronically infected HCV patients (*Jones et al., 2017*). Another RIG-I agonist, MK-4621, appears to be safe and well tolerable for patients with advanced or recurrent tumors, with no dose-limiting toxicities (*Middleton et al., 2018*).

The posttranscriptional modifications of RIG-I are vital for its activation and stability. Several E3 ligases have been reported to catalyze K63- or K48-linked polyubiquitination of RIG-I, regulating the RLR pathway. K63-linked ubiquitination, mediated by TRIM25 (*Gack et al., 2007*) and Mex-3 RNA binding family member C (MEX3C) in the CARD region of RIG-I (*Yan et al., 2014*; *Kuniyoshi et al., 2014*) or by RNF135 in the C-terminus of RIG-I (*Gao et al., 2009*; *Oshiumi et al., 2009*), facilitates RLR signal transduction. In addition to K63-linked ubiquitination that usually associates with signal transduction pathways, classical degradative K48-linked polyubiquitylation also regulates RIG-I's stability. Several E3 ligases such as c-Cbl (*Chen et al., 2013*), ring finger protein 122 (RNF122) (*Wang et al., 2016*), RNF125 (*Arimoto et al., 2007*), and TRIM40 (*Zhao et al., 2017*) catalyze this process. Conversely, several deubiquitylating enzymes also regulate RIG-I expression. For example, ubiquitin-specific peptidase 3 (USP3), USP21, and cylindromatosis lysine 63 deubiquitinase (CYLD) modulate RIG-I signaling by removing K63-polyubiquitin chains (*Cui et al., 2014*; *Fan et al., 2014*). Deubiquiti-nase USP4 and USP15 can increase the stability of RIG-I and TRIM25 by decreasing their K48-linked ubiquitination (*Wang et al., 2013*; *Pauli et al., 2014*). Discovering new proteins regulating the activity or stability of RIG-I will provide new insights and targets for antiviral and anti-tumor therapies.

ORMDL3 is a member of the three-gene ORDML family (ORMDL1, ORMDL2, and ORMDL3), and it is a 153aa transmembrane protein primarily located in the endoplasmic reticulum (ER) (*Hitomi et al., 2017*). Genetic variants in *ORMDL3* are associated with sphingolipid synthesis and altered metabolism, which contribute to asthma (*Zhang, 2021*). Recent evidence has elucidated that ORMDL3 regulates eosinophil trafficking, recruitment, and degranulation, which may induce the formation of allergic asthma and potentially other eosinophilic disorders (*Ha et al., 2013*). Additionally, ORMDL3 polymorphisms also contribute to a diverse set of inflammatory disorders that include bronchial asthma, inflammatory bowel disease (*McGovern et al., 2010*), ankylosing spondylitis (*Qiu et al., 2013*), atherosclerosis (*Ma et al., 2015*), SLE (*Kurreeman et al., 2012*), and cholangitis (*Hitomi et al., 2017*; *Xiang et al., 2021*). However, the role of ORMDL3 in innate immunity remains unknown.

In this study, we illuminate ORMDL3 as a pivotal negative regulator of the type I IFN signaling pathway. ORMDL3 forms a complex with MAVS and subsequently directs RIG-I toward degradation. ORMDL3 amplifies the K48-linked ubiquitination of RIG-I by disrupting the interaction between RIG-I and USP10. Animal experiments showed that inhibiting ORMDL3 enhances anti-tumor activity, demonstrated by an augmented proportion of activated CD8$^+$ T cells and increased interferon production within the tumor microenvironment (TME). Collectively, our results unveil the critical role of ORMDL3 in maintaining the homeostasis of antiviral innate immune responses and suggest ORMDL3 as a candidate target for cancer immunotherapy.

## Results

### ORMDL3 negatively regulates RLR-induced type I IFN signaling pathway

In order to investigate the potential role of ORMDL3 in the antiviral response, HEK293T cells over-expressing ORMDL3 were stimulated with poly(I:C) or vesicular stomatitis virus (VSV) infection. The result showed that ORMDL3 significantly inhibited poly(I:C) and VSV stimulated transcription of *IFNB1* (*Figure 1A*). Western blots demonstrated a marked reduction in the phosphorylation level of IRF3 when ORMDL3 was overexpressed (*Figure 1B*). Ectopic expression of ORMDL3 in A549 cells attenuated *IFNB1* expression induced by poly(I:C) but not poly(dG:dC) (*Figure 1C*). This phenomenon was also observed in parallel experiments using mouse bone marrow-derived primary macrophages (BMDM) (*Figure 1D*). These findings highlight ORMDL3 as a repressor of RNA-induced type I IFN expression. To unravel the molecular mechanism underlying the suppression of type I IFN signaling by ORMDL3, luciferase reporter assays were performed. ORMDL3 was found to decrease the IFNβ luciferase reporter activity induced by RIG-I while showing no effect on cGAS/STING or TRIF (*Figure 1E*). The ISRE luciferase reporter assay also showed similar results (*Figure 1F*). Furthermore, we constructed ORMDL3 stable knockdown and overexpression cell lines of A549 cells to examine the role of ORMDL3 on viral replication. We found that ORMDL3 knockdown strikingly suppressed the replication of VSV, whereas overexpression of ORMDL3 enhanced the replication of VSV (*Figure 1G*). We also infected shNC and shORMDL3 A549 stable cell lines with herpes simplex virus-1 (HSV-1) and found no significant difference in viral replication (*Figure 1H*). These results suggest that ORMDL3 only facilitates RNA virus replication but not DNA virus, which coincides with the finding that ORMDL3 specifically represses RNA- but not DNA-induced type I IFN expression (*Figure 1C, D*).

We next evaluated the influence of VSV, HSV-1, and RIG-I agonist SB9200 on ORMDL3 expression. Given the single nucleotide polymorphism (SNP) site rs7216389 at ORMDL3 locus is associated with the susceptibility of childhood asthma (*Moffatt et al., 2007*) and virus-induced respiratory wheezing illnesses (*Calışkan et al., 2013*), we took the genotype of this SNP into account. Upon these stimuli, HSV-1 does not obviously alter the abundance of ORMDL3 (*Figure 1—figure supplement 1A*). For VSV and SB9200, the expression of ORMDL3 is downregulated in some cell lines and is independent of the SNP (*Figure 1—figure supplement 1B and C*). Taken together, these results suggest that ORMDL3 is a negative regulator of RLR RNA sensing pathway, and its expression is reciprocally repressed by this pathway.

### ORMDL3 regulates the protein abundance of RIG-I

Further quantitative real-time PCR (qRT-PCR) results indicated that ectopic expression of ORMDL3 inhibited *IFNB1* mRNA expression and transcription of downstream genes *CCL5* and *CXCL10* induced by RIG-I and MAVS but not TBK1 or IRF3-5D (*Figure 2A and B*, *Figure 2—figure supplement 1A*). Additionally, MDA5-induced IFN upregulation was also inhibited by ORMDL3 (*Figure 2—figure supplement 1B*). These results revealed that ORMDL3 negatively regulates RLR pathway. Since ORMDL3 was downregulated in response to VSV stimulation in HEK293T, we first transfected cells with siORMDL3 followed by secondary transfection with RIG-I-N (an active form of RIG-I, 1–200aa of RIG-I) in HEK293T, and we found that *ORMDL3* knockdown significantly increased the expression of *IFNB1*, *CCL5*, and *ISG54* (*Figure 2C*) as well as the protein abundance of RIG-I (*Figure 2D*). As the ORMDL3 antibody can recognize all ORMDL family members (ORMDL1, 2, and 3), we also detected the mRNA level of *ORMDL3* to further validate its knockdown efficiency (*Figure 2—figure supplement 1C*). Subsequent experiments involving various signaling proteins such as RIG-I (WT/ RIG-I-N), MDA5, TBK1, and IRF3 indicated a negative correlation between ORMDL3 levels and RIG-I/ RIG-I-N protein expression, with maximal degradation observed in RIG-I-N (*Figure 2E*). To investigate whether ORMDL3 is unique in promoting RIG-I degradation, we compared ORMDL1, 2, and 3. When we co-expressed RIG-I-N with them, we found that only ORMDL3 can facilitate the degradation of RIG-I-N (*Figure 2—figure supplement 1D*). In addition, we also tested whether overexpressing murine Ormdl3 will lead to murine Rig-I degradation. Interestingly, we discovered that human RIG-I and murine Rig-I can be degraded upon ORMDL3/Ormdl3 overexpression, implying that ORMDL3's function is conservative in humans and mice (*Figure 2—figure supplement 1E and F*). In addition, ORMDL3 overexpression also eliminated endogenous RIG-I protein abundance (*Figure 2—figure supplement 1G*).

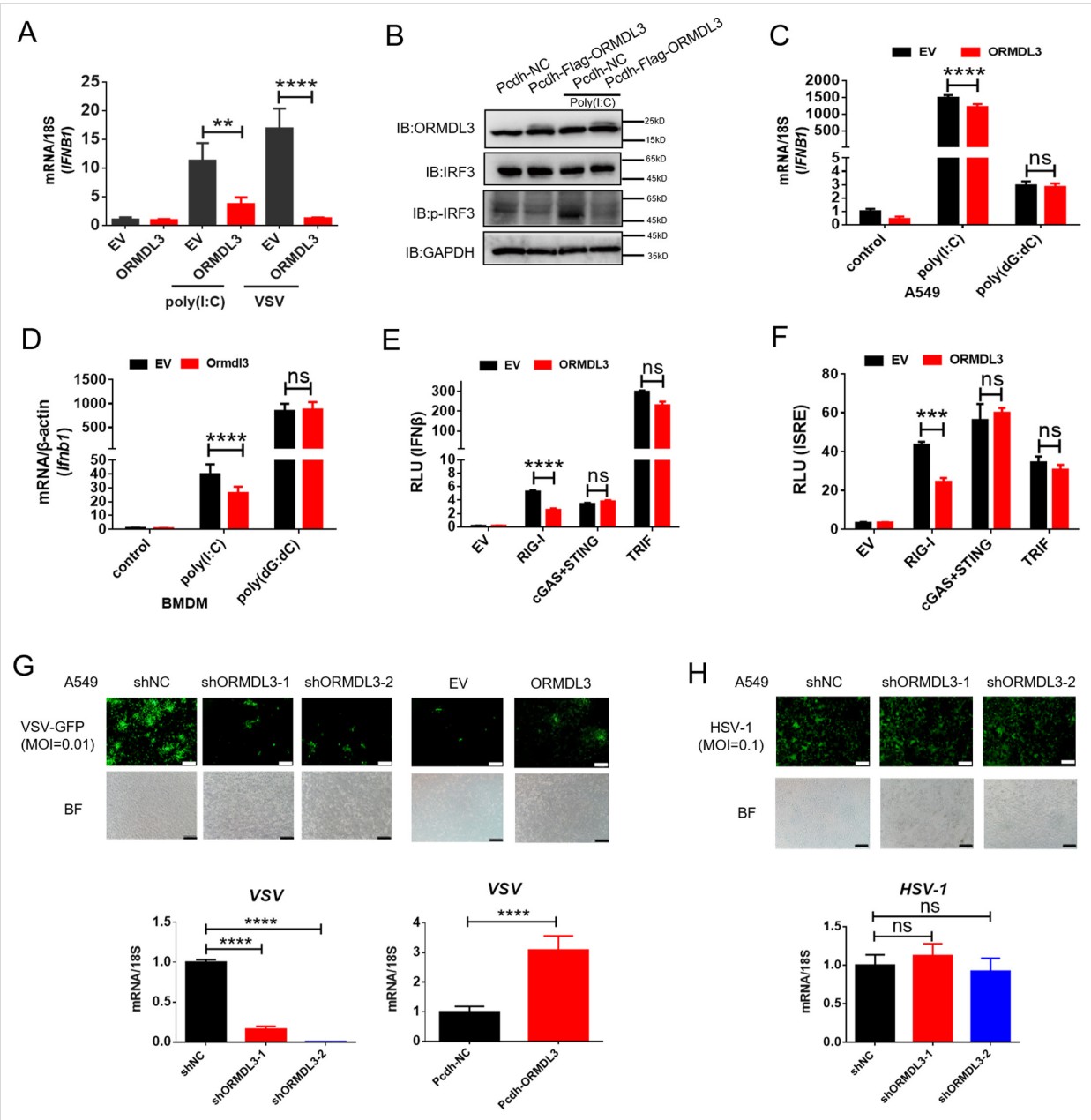

**Figure 1.** ORMDL3 negatively regulates RIG-I-like receptor (RLR)-induced type I interferon (type I IFN) signaling pathway. (A) HEK293T cells were transfected with an empty vector (EV) or ORMDL3 plasmid for 12 hr and were then infected with vesicular stomatitis virus (VSV) (MOI = 0.01) or transfected with poly(I:C). The transcription of *IFNB1* mRNA was detected using quantitative real-time PCR (qRT-PCR). (B) HEK293T-EV and HEK293T-Flag-ORMDL3 stable cell lines were transfected with or without poly(I:C) and immunoblot analyses of phosphorylated IRF3 (p-IRF3), total IRF3, GAPDH, and ORMDL3 levels were performed. (C) Results of the qRT-PCR assays showing mRNA levels of *IFNB1* in A549 cells transfected with EV or ORMDL3 followed by stimulating with poly(I:C) or poly(dG:dC). (D) Results of the qRT-PCR assays showing mRNA levels of *Ifnb1* in bone marrow-derived primary macrophages (BMDM) infected with EV or psc-AAV-Ormdl3 virus followed by transfecting with poly(I:C) or poly(dG:dC). (E–F) Results of the luciferase assay showing IFNβ-Luc activity (E) and ISRE-Luc activity (F) in HEK293T cells transfected with EV or ORMDL3 plasmids together with individual EV, RIG-I, cGAS plus STING, or TRIF plasmids for 24 hr. (G) ORMDL3 stable knockdown or overexpression A549 cells, and the control cells were infected with VSV-GFP (MOI = 0.01) for 12 hr. The viral infection was observed using fluorescence microscopy, and the viral amount was detected using qRT-PCR. Scale bars, 200 μm. (H) The control and ORMDL3 stable knockdown A549 cells were infected with herpes simplex virus-1 (HSV-1) (MOI = 0.1) for 24 hr. The viral infection was observed using fluorescence microscopy, and the viral amount was detected using qRT-PCR. Scale bars, 200 μm. Data from three independent experiments are presented as mean ± SD and were analyzed by two-tailed Student's t test (A, C, D-F, G-H, bottom) ,**p<0.01, ***p<0.001, ****p<0.0001, and ns = no significance.

The online version of this article includes the following source data and figure supplement(s) for figure 1:

*Figure 1 continued on next page*

*Figure 1 continued*

**Source data 1.** Original files for western blots shown in *Figure 1*, indicating relevant bands.

**Source data 2.** Original files for western blots shown in *Figure 1*.

**Figure supplement 1.** ORMDL3 expression under different treatments.

**Figure supplement 1—source data 1.** Original files for western blots shown in *Figure 1—figure supplement 1*, indicating relevant bands.

**Figure supplement 1—source data 2.** Original files for western blots shown in *Figure 1—figure supplement 1*.

Based on these observations, we focus on the relationship between RIG-I and ORMDL3. We ectopically expressed an increasing amount of Flag-ORMDL3 with the RIG-I-N and the RIG-I-C truncated form (*Figure 2—figure supplement 1H*) and found that ORMDL3 only decreased the protein abundance of RIG-I-N (*Figure 2F*). We co-transfected ORMDL3 and RIG-I with or without poly(I:C) and found that RIG-I was degraded upon the stimulation of poly(I:C), suggesting that ORMDL3 degrades RIG-I only when it was activated and the CARD was exposed (*Figure 2G*). Further examination showed that co-expression of ORMDL3 suppressed RIG-I-N-induced expression of *IFNB1* and *CCL5*, as well as pro-inflammatory cytokines *IL6* and *TNF* (*Figure 2H*), suggesting its role on both NF-κB and type I IFN pathways. Given the regulation of ORMDL3 on NF-κB has been reported (*Bugajev et al., 2016*), we focused on its role in the type I IFN pathway.

## ORMDL3 promotes proteasomal degradation of RIG-I

There are two major protein degradation pathways, including the ubiquitin-proteasome pathway and the lysosomal proteolysis system. We next identified which degradation system dominantly mediates the degradation of RIG-I by ORMDL3. We co-expressed ORMDL3 and RIG-I-N and treated cells with proteasome inhibitor MG132, and lysosome inhibitor CQ. We found the degradation of RIG-I-N mediated by ORMDL3 could be blocked by MG132 but not CQ (*Figure 3A*). To rule out the possibility of transcriptional downregulation of RIG-I, qRT-PCR analysis was performed, confirming that the decrease in RIG-I protein was a post-transcriptional event (*Figure 3—figure supplement 1A*). To investigate the mechanism, we co-transfected RIG-I-N, ORMDL3, and plasmids encoding different forms of ubiquitin. The results indicated an increase in K48-linked ubiquitination on RIG-I-N, implicating ORMDL3 in promoting proteasomal degradation of RIG-I (*Figure 3B*). To pinpoint the lysine residues crucial for RIG-I ubiquitination, we engineered a mutant version of RIG-I-N in which all lysines were mutated to arginines, denoted as RIG-I-N-KR. Intriguingly, the degradation-promoting effect of ORMDL3 on RIG-I-N-KR was nullified (*Figure 3C*). Given there are 18 lysine residues in RIG-I-N, we generated two mutants, mutant1 and mutant2, each mutating the last nine lysines and the first nine lysines, respectively (*Figure 3—figure supplement 1B*). Remarkably, the results showed that mutant1 was resistant to degradation by ORMDL3 (*Figure 3D*), suggesting that the last nine lysines on RIG-I-N may mediate its degradation induced by ORMDL3.

To delve deeper into the intricate mechanism of ORMDL3-induced degradation of RIG-I, we initially introduced single-point mutations, specifically K146R, K154R, K164R, and K172R (*Rehwinkel and Gack, 2020*), which have been reported important for the function and stability of RIG-I. Co-transfection with ORMDL3 revealed that these individual mutations did not impede the degradation process, hinting at the potential cooperation of lysine residues (*Figure 3E*). We then mutated all four lysine residues and assessed ORMDL3-induced RIG-I-N degradation. Strikingly, the RIG-I-N-4KR mutant, in which K146, K154, K164, and K172 were simultaneously mutated to arginines, displayed resistance to degradation by ORMDL3 (*Figure 3F*). At the meantime, the RIG-I-N 4KR mutant failed to exhibit the upregulation of K48-linked ubiquitination induced by ORMDL3 overexpression, reinforcing the pivotal role played by K146, K154, K164, and K172 in mediating RIG-I ubiquitination and subsequent degradation (*Figure 3G*). In addition, we found that when Ormdl3 was overexpressed in BMDM, endogenous Rig-I was downregulated (*Figure 3H*).

## ORMDL3 interacts with the signaling adaptor MAVS

Next, we sought to determine the binding partner of ORMDL3 in the type I IFN pathway. Co-immunoprecipitation (co-IP) and immunoblot analysis showed that only Flag-tagged MAVS interacted with ORMDL3-GFP (*Figure 4A*). To delineate the requisite domains of MAVS facilitating this interaction, various MAVS truncations were co-transfected with ORMDL3. Notably, deletion of the

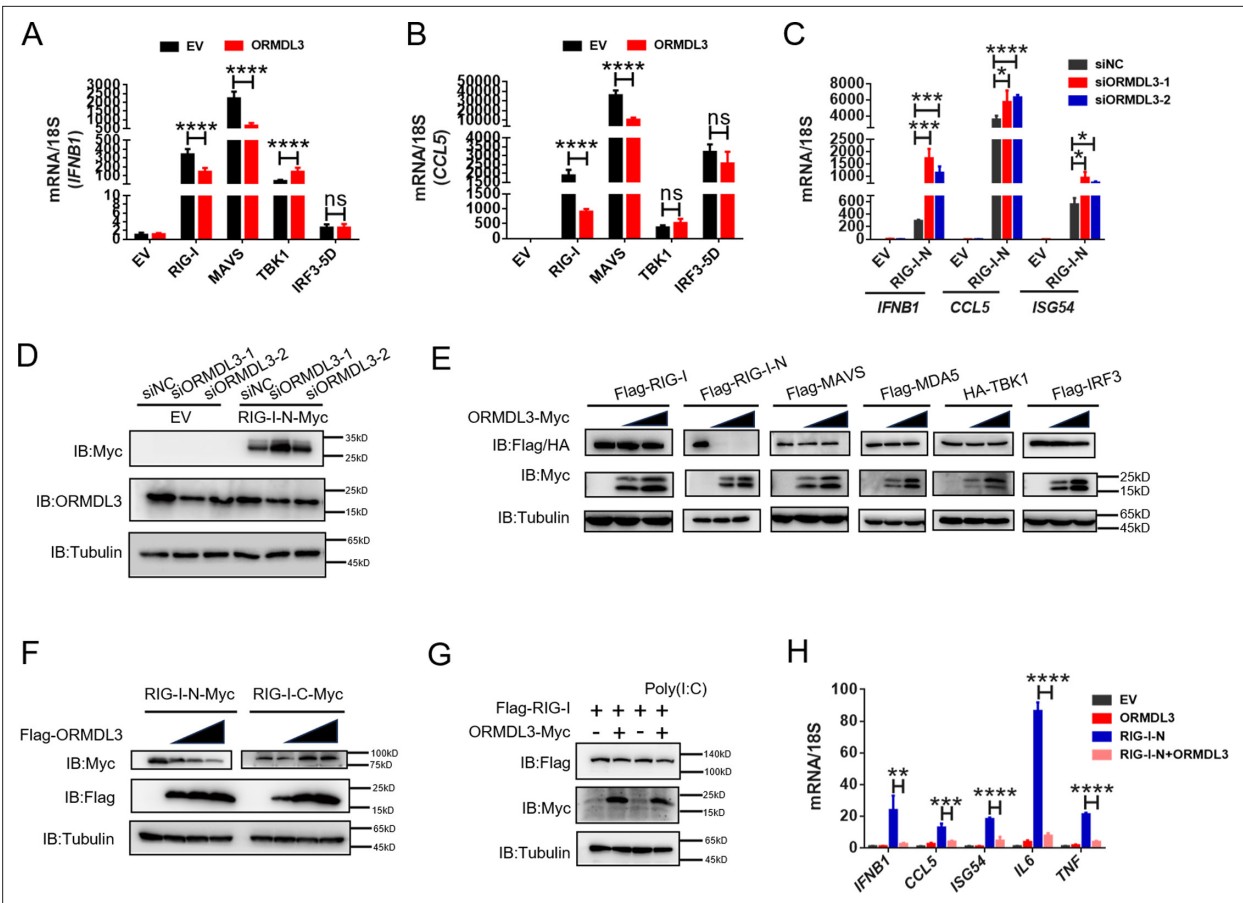

**Figure 2.** ORMDL3 regulates the protein abundance of RIG-I. (A, B) Quantitative real-time PCR (qRT-PCR) analyses of the expression of the indicated mRNA levels in HEK293T cells transfected with empty vector (EV) or ORMDL3 plasmids combined with co-transfection of individual plasmids encoding EV, RIG-I, MAVS, TBK1, IRF3-5D (the constitutively activated form of IRF3). (C) Results of the qRT-PCR assays showing mRNA levels of *IFNB1*, *CCL5*, and *ISG54* in HEK293T cells transfected with control or ORMDL3-specific siRNAs followed by secondary transfection with EV or RIG-I-N plasmids. (D) Results of the WB assays showing protein levels in HEK293T cells transfected with control or ORMDL3-specific siRNAs followed by secondary transfection with EV or RIG-I-N plasmids. (E) Immunoblot analysis of 293T cells transfected with individual plasmid encoding Flag-tagged RIG-I, RIG-I-N, MAVS, MDA5, IRF3, or HA-TBK1, in combination with increasing doses of ORMDL3-Myc plasmids. (F) Immunoblot analysis of 293T cells transfected with plasmid of RIG-I-N-Myc or RIG-I-C-Myc and increasing doses of Flag-ORMDL3 plasmid. (G) HEK293T cells were transfected with Flag-RIG-I and ORMDL3-Myc plasmids, as indicated, with or without poly(I:C) co-transfection. Cell lysates were immunoblotted with anti-Flag and anti-Myc antibodies. (H) Results of the qRT-PCR assays showing *IFNB1*, *CCL5*, *ISG54*, *IL6*, and *TNF* mRNA levels in HEK293T cells transfected with EV, ORMDL3, and RIG-I-N plasmids as indicated for 24 hr. Data from three independent experiments are presented as mean ± SD and were analyzed by two-tailed Student's t test (A-C, H), *p<0.05, **p<0.01, ***p<0.001, ****p<0.0001, and ns = no significance.

The online version of this article includes the following source data and figure supplement(s) for figure 2:

**Source data 1.** Original files for western blots shown in *Figure 2*, indicating relevant bands.

**Source data 2.** Original files for western blots shown in *Figure 2*.

**Figure supplement 1.** ORMDL3-mediated downregulation of RIG-I is conserved in both human and murine cells.

**Figure supplement 1—source data 1.** Original files for western blots shown in *Figure 2—figure supplement 1*, indicating relevant bands.

**Figure supplement 1—source data 2.** Original files for western blots shown in *Figure 2—figure supplement 1*.

transmembrane domain (TM) of MAVS abrogated the interaction, while deletion of the CARD had no discernible impact (*Figure 4B*, *Figure 4—figure supplement 1A*). It has been reported that ORMDL3 contains four TM segments (TM1-TM4) (*Li et al., 2021*). To map the essential domains of ORMDL3 that mediate its association with MAVS, we generated four truncations of ORMDL3 based on the structure, which are 1–42aa, 43–82aa, 83–118aa, and 119–153aa (*Figure 4—figure supplement 1B*, upper panel). Intriguingly, we found that all these four ORMDL3 truncations interact with MAVS (*Figure 4C*) and impede RIG-I-N-induced transcription of *IFNB1*, *CCL5*, *ISG54*, and *ISG56* (*Figure 4D*).

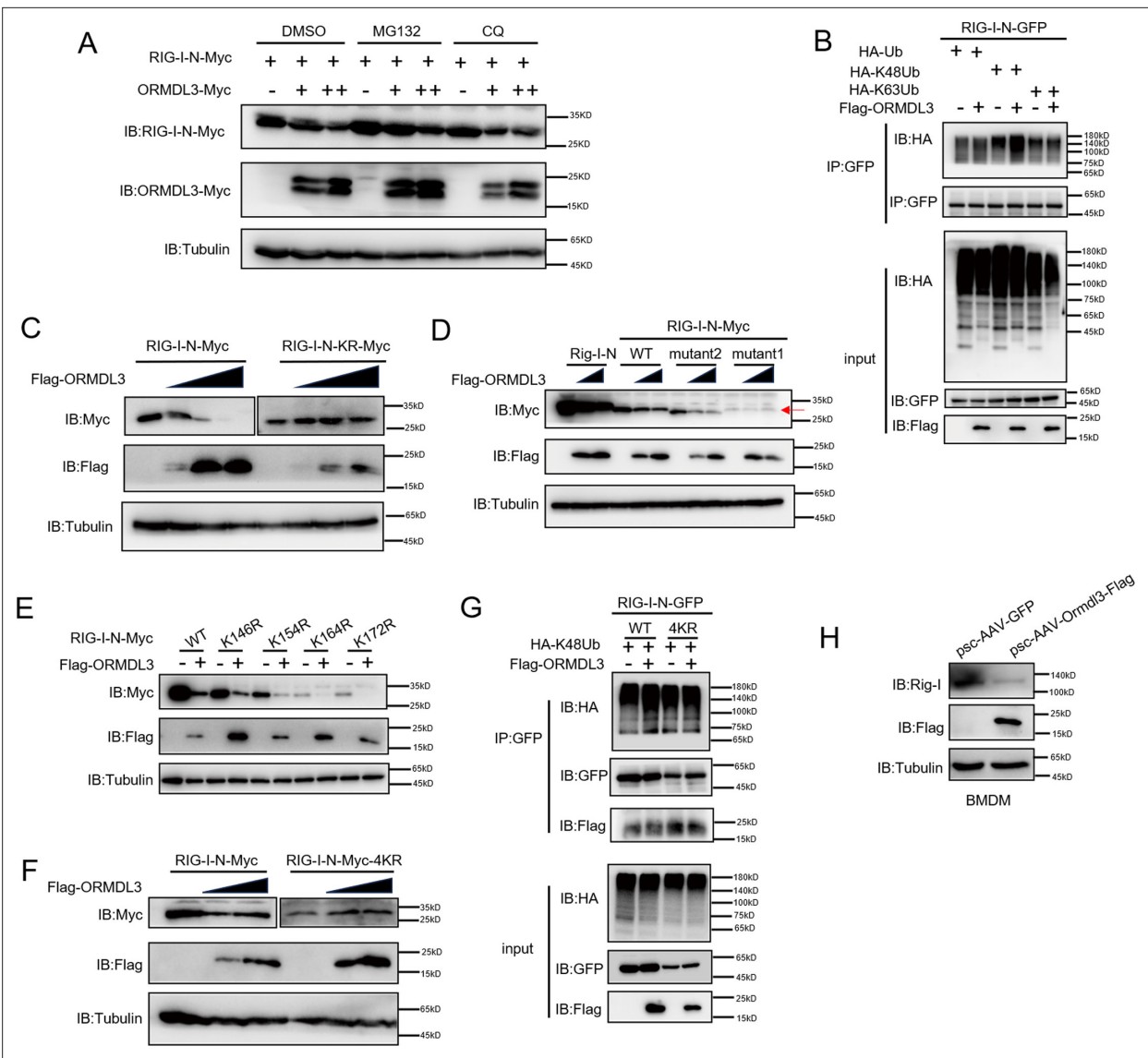

**Figure 3.** ORMDL3 promotes proteasomal degradation of RIG-I. (A) HEK293T cells were transfected with plasmids encoding RIG-I-N-Myc together with increasing amounts of Flag-ORMDL3 plasmid treated with MG132 (10 μM) or chloroquine (CQ) (50 μM) for 6 hr and the cell lysates were analyzed by immunoblot. (B) HEK293T cells were transfected with the indicated plasmids, and cell lysates were immunoprecipitated with an anti-GFP antibody followed by immunoblots using anti-GFP and anti-HA antibodies. (C) HEK293T cells were transfected with RIG-I-N-Myc (WT or KR) and increasing doses of plasmid for Flag-ORMDL3. The expression levels of RIG-I-N-Myc were analyzed by immunoblot. (D) HEK293T cells were transfected with Rig-I-N, RIG-I-N-Myc (WT, KR, mutant1, or mutant2) and increasing doses of plasmid for Flag-ORMDL3. The expression levels of RIG-I-N-Myc were analyzed by immunoblot. (E) 293T cells were transfected with RIG-I-N-Myc (WT, K146R, K154R, K164R, or K172R) with or without Flag-ORMDL3. The expression levels of RIG-I-N-Myc and its mutant forms were analyzed by immunoblot. (F) HEK293T cells were transfected with RIG-I-N-Myc (WT or 4KR) and increasing doses of Flag-ORMDL3. The expression levels of RIG-I-N-Myc (WT or 4KR) were analyzed by immunoblot. (G) HEK293T cells were transfected with RIG-I-N-GFP (WT or 4KR) and HA-K48Ub in combination with EV or Flag-ORMDL3, and cell lysates were immunoprecipitated with an anti-GFP antibody followed by immunoblots using anti-GFP, anti-HA, and anti-Flag antibodies. (H) Bone marrow-derived primary macrophages (BMDM) were infected with psc-AAV-GFP or psc-AAV-Ormdl3-Flag virus, followed by immunoblot analysis of Rig-I, Flag, and Tubulin.

The online version of this article includes the following source data and figure supplement(s) for figure 3:

**Source data 1.** Original files for western blots shown in *Figure 3*, indicating relevant bands.

**Source data 2.** Original files for western blots shown in *Figure 3*.

**Figure supplement 1.** ORMDL3 promotes the degradation of RIG-I.

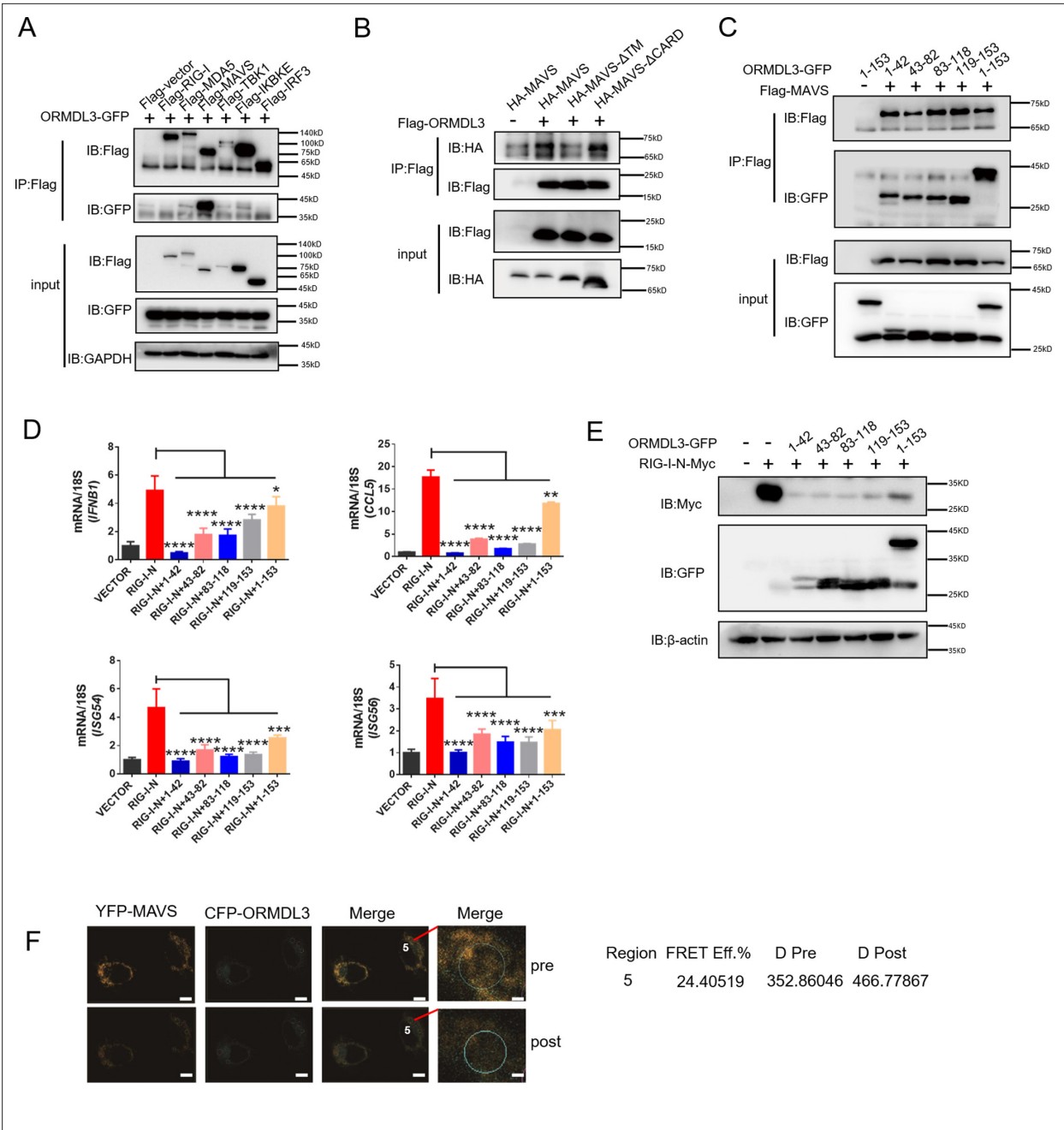

**Figure 4.** ORMDL3 interacts with signaling adaptor MAVS. (A) HEK293T cells were transfected with empty vector (EV) or Flag-RIG-I/MAVS/MDA5/ TBK1/IRF3/IKBKE with ORMDL3-GFP, and cell lysates were immunoprecipitated with an anti-Flag antibody followed by immunoblots using anti-GFP and anti-Flag antibodies. (B) HEK293T cells were transfected with different MAVS truncations in combination with EV or Flag-ORMDL3, and cell lysates were immunoprecipitated with anti-Flag antibody followed by immunoblots using anti-HA and anti-Flag antibodies. (C) HEK293T cells were transfected with EV or Flag-MAVS in combination with different ORMDL3 truncations, and cell lysates were immunoprecipitated with anti-Flag antibody followed by immunoblots using anti-GFP and anti-Flag antibodies. (D) HEK293T cells were transfected with EV or Flag-MAVS in combination with different ORMDL3 truncations, followed by quantitative real-time PCR (qRT-PCR) analysis of *IFNB1*, *CCL5*, *ISG54*, and *ISG56*. (E) HEK293T cells were transfected with RIG-I-N-Myc in combination with EV or different ORMDL3 truncations, followed by immunoblots using anti-GFP and anti-Myc antibodies. (F) FRET experiment of YFP-MAVS and CFP-ORMDL3 in HeLa cells. YFP-MAVS is the donor and CFP-ORMDL3 is the acceptor, and FRET efficiency is 24.40519%. Scale bars, 10 μm. Data from three independent experiments are presented as mean ± SD and were analyzed by two-tailed Student's t test (D), *p<0.05, **p<0.01, ***p<0.001, ****p<0.0001.

The online version of this article includes the following source data and figure supplement(s) for figure 4:

**Source data 1.** Original files for western blots shown in *Figure 4*, indicating relevant bands.

*Figure 4 continued on next page*

*Figure 4 continued*

**Source data 2.** Original files for western blots shown in *Figure 4*.

**Figure supplement 1.** All truncations of ORMDL3 can inhibit type I interferon (type I IFN) production.

**Figure supplement 1—source data 1.** Original files for western blots shown in *Figure 4—figure supplement 1*, indicating relevant bands.

**Figure supplement 1—source data 2.** Original files for western blots shown in *Figure 4—figure supplement 1*.

In addition, co-transfection of RIG-I-N-Myc with individual ORMDL3 truncations showed that each domain of ORMDL3 is favorable to RIG-I-N degradation (*Figure 4E*). Moreover, inspired by Guo et al.'s discovery of a naturally occurring short isoform of ORMDL3 (*Jin et al., 2011*), we engineered N- and C-terminal truncations of ORMDL3 to mimic this isoform (*Figure 4—figure supplement 1B*, lower panel), and the results showed that both truncations of ORMDL3 retained the ability to interact with MAVS (*Figure 4—figure supplement 1C*). Luciferase assays and qRT-PCR assays confirmed the inhibitory efficacy of each domain of ORMDL3 (*Figure 4—figure supplement 1D and E*). We also tested whether these different domains could promote the degradation of RIG-I-N, and the results revealed that both domains of ORMDL3 could enhance the degradation of RIG-I (*Figure 4—figure supplement 1F*). To further validate the association between ORMDL3 and MAVS, we performed a FRET experiment in HeLa cells. In this experiment, we co-overexpressed YFP-MAVS (donor) and CFP-ORMDL3 (acceptor). When we bleached YFP-MAVS, we noticed that the fluorescence of CFP-ORMDL3 increased, indicating a direct interaction between ORMDL3 and MAVS (*Figure 4F*).

## USP10 deubiquitinates and stabilizes RIG-I

To identify the E3 ligase or deubiquitinase involved in ORMDL3-mediated RIG-I ubiquitination, we conducted immunoprecipitation-mass spectrometry (IP-MS) analysis using Flag-ORMDL3 as bait and identified USP10 as a potential candidate (*Figure 5A*). We validated the IP-MS results and the co-IP experiment revealed that only USP10 can interact with ORMDL3 but not CAND1 or UFL1 (*Figure 5—figure supplement 1*). Subsequent co-IP validation demonstrated the interaction between ORMDL3 and USP10, USP10 and RIG-I, respectively (*Figure 5B and C*). Interestingly, we found that the RIG-I level is decreased in USP10 stable knockdown HEK293T cells while overexpression of USP10 promotes the accumulation of RIG-I (*Figure 5D and E*). As USP10 is a deubiquitinase, we investigated its impact on RIG-I ubiquitination and observed a decrease in K48-linked ubiquitination of RIG-I upon USP10 overexpression (*Figure 5F*). Co-transfection USP10 with RIG-I-N or its 4KR mutant showed that USP10 failed to increase the RIG-I-N-4KR level, underscoring the indispensability of these four lysine residues (*Figure 5G*). Upon overexpressing RIG-I-N and ORMDL3 in USP10 knockdown cells, ORMDL3's ability to degrade RIG-I-N was markedly compromised, emphasizing the indispensable role of USP10 in this degradation process (*Figure 5H*). These results further support that ORMDL3 interferes with USP10's function in regulating RIG-I.

## ORMDL3 disturbs USP10-induced RIG-I stabilization

Co-IP experiments unveiled robust binding of USP10 to both RIG-I and ORMDL3, and ORMDL3 disrupts the interaction between RIG-I and USP10 (*Figure 6A*). Crucially, USP10 exhibited a specific role in stabilizing RIG-I, but not other innate immune proteins such as MAVS, MDA5, and IRF3, and this effect can be reversed by ORMDL3 (*Figure 6B*). Further investigations showed that the function of ORMDL3 in disturbing USP10-mediated RIG-I stabilization could be rescued by the proteasome inhibitor MG132 but not the lysosome inhibitor CQ (*Figure 6C*). Subsequent co-transfection experiments delineated that RIG-I-N-KR or RIG-I-N-mutant1 was necessary to prevent USP10-induced accumulation and ORMDL3-mediated degradation (*Figure 6D*). Notably, single-point mutation of the K146, K154, K164, or K172 residue on RIG-I-N does not affect the regulation of USP10 and ORMDL3 (*Figure 6E*), whereas the RIG-I-N-4KR mutation abolished this process (*Figure 6F*). These findings underscored the importance of these four lysine residues in both ORMDL3-mediated degradation and USP10-mediated stabilization of RIG-I. Building upon these observations, we sought to elucidate whether USP10 influences RIG-I ubiquitination through these four sites. Co-expression of HA-K48Ub and RIG-I-N-GFP with or without USP10 revealed a decrease in K48-linked ubiquitination of RIG-I-N by USP10 and this effect was nullified in the presence of the 4KR mutant, which is in consistent with ORMDL3-mediated regulation (*Figure 6G*). Additionally, we verified the functional consequences of

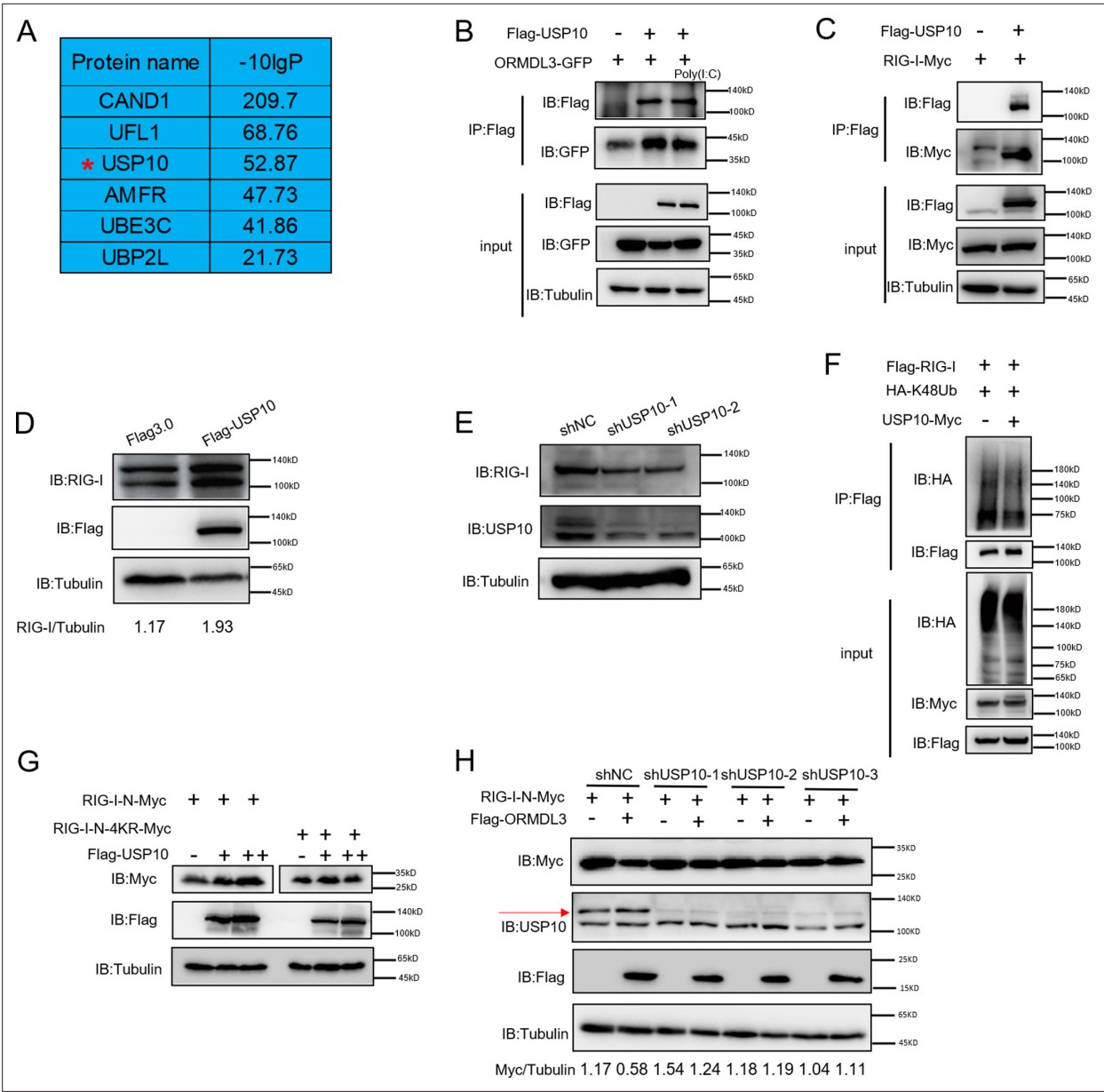

**Figure 5.** USP10 induces RIG-I stabilization. (A) Candidate proteins interacted with ORMDL3 screened from mass spectrometry results. (B) HEK293T cells were transfected with ORMDL3-GFP and Flag-USP10 plasmids, as indicated, with or without poly(I:C) co-transfection. Cell lysates were immunoprecipitated with the anti-Flag antibody and immunoblotted with anti-Flag and anti-GFP antibodies. (C) HEK293T cells were transfected with RIG-I-Myc and EV or Flag-USP10 plasmids. Cell lysates were immunoprecipitated with the anti-Flag antibody and immunoblotted with anti-Flag and anti-Myc antibodies. (D) Immunoblot the protein level of RIG-I in USP10 stable overexpression HEK293T cell line. (E) Immunoblot the protein level of RIG-I in USP10 stable knockdown HEK293T cell lines. (F) Immunoprecipitation (IP) and immunoblot analysis of 293T cells transfected with vectors expressing Flag-RIG-I and HA-K48Ub with or without USP10-Myc. (G) HEK293T cells were transfected with RIG-I-N-Myc (WT or 4KR) and increasing doses of expression vector for Flag-USP10. The expression levels of RIG-I-N-Myc were analyzed by immunoblot. (H) USP10 stable knockdown HEK293T cell lines were transfected with RIG-I-N-Myc and Flag-ORMDL3 as indicated. The expression levels of RIG-I-N-Myc were analyzed by immunoblot.

The online version of this article includes the following source data and figure supplement(s) for figure 5:

**Source data 1.** Original files for western blots shown in *Figure 5*, indicating relevant bands.

**Source data 2.** Original files for western blots shown in *Figure 5*.

**Figure supplement 1.** ORMDL3 interacts with USP10 but not CAND1 or UFL1.

**Figure supplement 1—source data 1.** Original files for western blots shown in *Figure 5—figure supplement 1*, indicating relevant bands.

**Figure supplement 1—source data 2.** Original files for western blots shown in *Figure 5—figure supplement 1*.

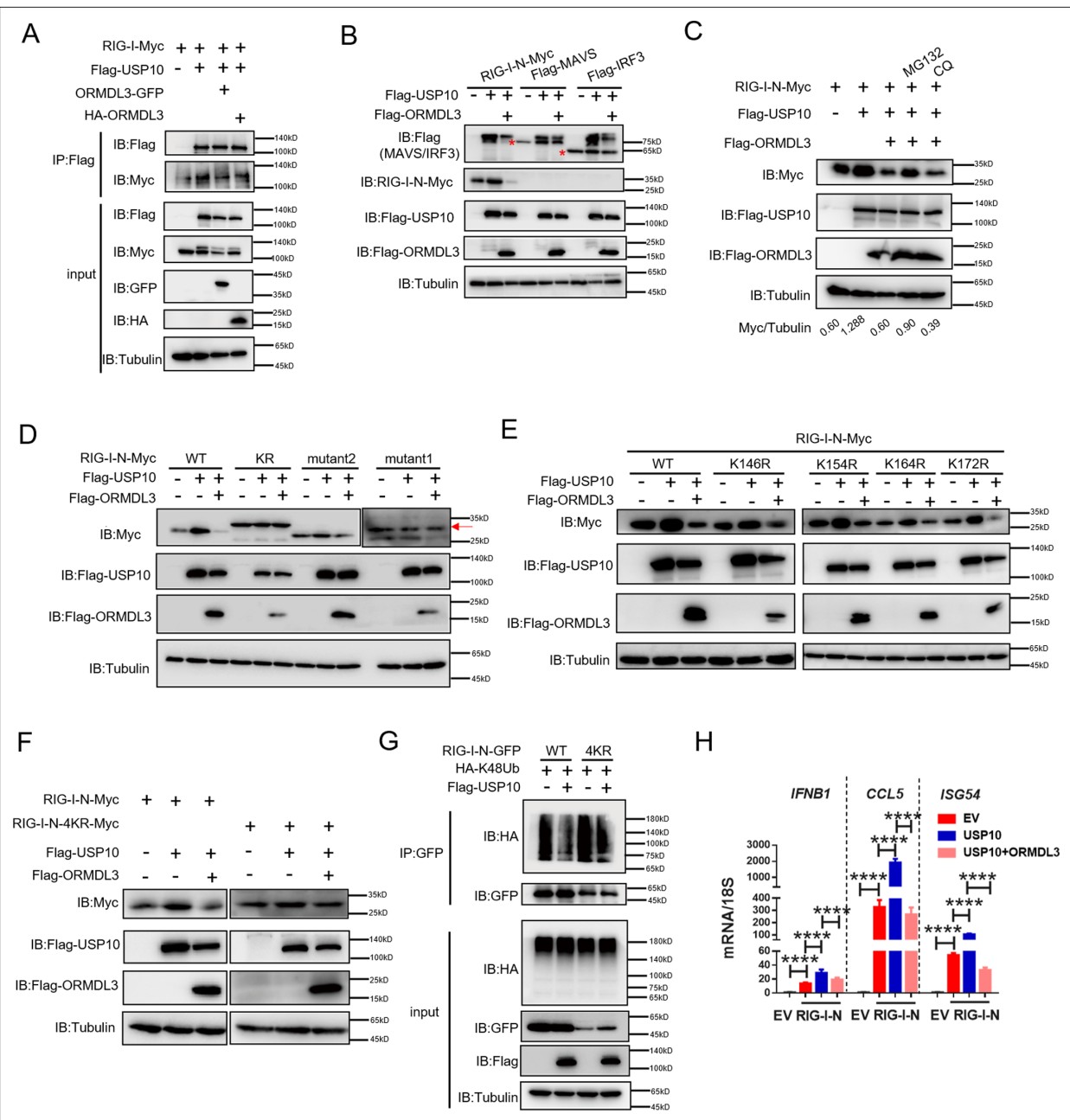

**Figure 6.** ORMDL3 disturbs USP10-induced RIG-I stabilization. (A) HEK293T cells were transfected with Flag-USP10 and RIG-I-Myc plasmids, as indicated, with or without ORMDL3 co-transfection. Cell lysates were immunoprecipitated with the anti-Flag antibody and immunoblotted with anti-Flag and anti-Myc antibodies. (B) Immunoblot analysis of HEK293T cells transfected with Flag-USP10 and Flag-MAVS, Flag-IRF3, or RIG-I-N-Myc with or without ORMDL3 co-transfection using indicated antibodies. (C) HEK293T cells were transfected with plasmids encoding RIG-I-N-Myc together with Flag-USP10 with or without Flag-ORMDL3 plasmid followed by MG132 (10 μM) or chloroquine (CQ) (50 μM) treatment for 6 hr. The cell lysates were analyzed by immunoblot. (D) HEK293T cells were transfected with Flag-USP10 and RIG-I-N-Myc (WT, KR, mutant1, or mutant2) plasmids, as indicated, with or without Flag-ORMDL3 co-transfection. Cell lysates were immunoblotted with indicated antibodies. (E) HEK293T cells were transfected with RIG-I-N-Myc (WT, K146R, K154R, K164R, or K172R) and Flag-USP10 with or without Flag-ORMDL3. Cell lysates were immunoblotted with indicated antibodies. (F) HEK293T cells were transfected with Flag-USP10 and RIG-I-N-Myc (WT or 4KR) plasmids, as indicated, with or without Flag-ORMDL3 co-transfection. Cell lysates were immunoblotted with indicated antibodies. (G) Immunoprecipitation (IP) and immunoblot analysis of HEK293T cells transfected with vectors expressing RIG-I-N-GFP/RIG-I-N-4KR-GFP and HA-K48Ub with or without USP10 transfection. (H) HEK293T cells were transfected with Flag-USP10 and RIG-I-N-Myc, with or without ORMDL3 co-transfection followed by quantitative real-time PCR (qRT-PCR) analysis of *IFNB1*, *CCL5*, and *ISG54*. Data from three independent experiments are presented as mean ± SD and were analyzed by two-tailed Student's t test (H), ****p<0.0001.

The online version of this article includes the following source data for figure 6:

*Figure 6 continued on next page*

*Figure 6 continued*
**Source data 1.** Original files for western blots shown in *Figure 6*, indicating relevant bands.
**Source data 2.** Original files for western blots shown in *Figure 6*.

USP10-induced RIG-I stabilization by assessing the mRNA levels of *IFNB1, CCL5,* and *ISG54*, which were increased upon enforced USP10 expression and reduced upon co-expression with ORMDL3 (*Figure 6H*).

## Knocking down of ORMDL3 enhances anti-tumor immunity

To assess the impact of ORMDL3 on anti-tumor activity, we conducted knockdown experiments targeting ORMDL3 in LLC and MC38 murine cancer cell lines, followed by subcutaneous inoculation into C57BL/6 mice. The deficiency of ORMDL3 significantly suppressed LCC tumor growth compared to the control group (*Figure 7A–C*). This tumor growth inhibition by targeting ORMDL3 was further validated in the MC38 cancer model (*Figure 7G–I*). Moreover, in both LLC and MC38 knockdown cell lines, the protein level of Rig-I was significantly upregulated (*Figure 7—figure supplement 1A–D*), and its upregulation was further verified by western blots in LLC tumors (*Figure 7—figure supplement 1E*) and immunohistochemistry (IHC) in MC38 tumors (*Figure 7K*). Moreover, investigations revealed that in the LLC tumor model, the knockdown of ORMDL3 led to a significant increase in the expression of ISGs, including *Ccl5, Cxcl10, Tnf*, and *Il6*, compared to the control group (*Figure 7D*). This upregulation of ISGs upon ORMDL3 knockdown was consistent in the MC38 cancer model, where *Ifnb1*, *Ccl5*, and *Cxcl10* mRNA levels were significantly elevated (*Figure 7J*). Flow cytometry analysis demonstrated an increase in CD3$^+$ T cell infiltration percentage in LLC tumors when ORMDL3 was knocked down (*Figure 7E*). Notably, although CD8$^+$ cell levels showed no significant change among groups (*Figure 7—figure supplement 1F*), activated CD8$^+$ T cells (CD8$^+$ CD107a$^+$ and CD8$^+$ CD44$^+$) exhibited an increase in the ORMDL3 knockdown group (*Figure 7F*, *Figure 7—figure supplement 1G*). In addition, IHC assays revealed that more CD8$^+$ T cells were infiltrated in ORMDL3 knockdown MC38 tumors (*Figure 7K*). Collectively, these findings suggest that inhibition of tumor-intrinsic ORMDL3 amplifies anti-tumor immunity by increasing ISGs expression in TME and promoting cytotoxic CD8$^+$ T cell activation.

We further analyzed ORMDL3 expression in the TCGA-pan-cancers cohort. We observed higher expression of ORMDL3 in lung adenocarcinoma (LUAD), colon adenocarcinoma, and lung squamous cell carcinoma compared to their corresponding normal samples (*Figure 7—figure supplement 2A*). In LUAD cohorts, high ORMDL3 expression was associated with poor prognosis, as indicated by overall survival, progression-free survival, and disease-specific survival analyses (*Figure 7—figure supplement 2B–D*). Analysis of LUAD cohorts also revealed enrichment of stromal scores in tumors with low ORMDL3 expression (*Figure 7—figure supplement 2E*). Additionally, we found a negative correlation between ORMDL3 expression and the ESTIMATE score, indicating a potential association between ORMDL3 expression and immune cell infiltration (*Figure 7—figure supplement 2E*). Interestingly, ORMDL3 expression showed a negative correlation with CD8$^+$ T cell activation markers such as PRF1, GZMA, and GZMB, as well as with ISGs such as CCL5 and CXCL10 (*Figure 7—figure supplement 2F*), validating our finding that ORMDL3 serves as a negative regulator of the IFN signaling pathway and anti-tumor immunity.

## Discussion

The RIG-I MAVS pathway is essential for the early detection of viral infections and the initiation of an effective antiviral immune response. This pathway has two major downstream signaling events: the interferon signaling and the pro-inflammatory cytokine signal axis (*Wang et al., 2022*). During RNA virus infection, MAVS recruits TBK1, which phosphorylates IRF3 and IRF7, leading to the production of type I IFN. Meanwhile, the pro-inflammatory cytokine signaling pathway primarily operates through the activation of NF-κB, resulting in the production of pro-inflammatory cytokines, such as IL6, TNF, etc. (*Fitzgerald et al., 2003*). The activation of RIG-I is critical for innate immunity, and its post-translational modifications play a vital role. For instance, E3 ligases such as TRIM25, TRIM4, RNF135, and RNF194 facilitate RIG-I activation by mediating its K63-linked ubiquitination. Conversely, other E3 ligases, including RNF125, RNF122, casitas B-lineage lymphoma proto-oncogene (c-Cbl, also known

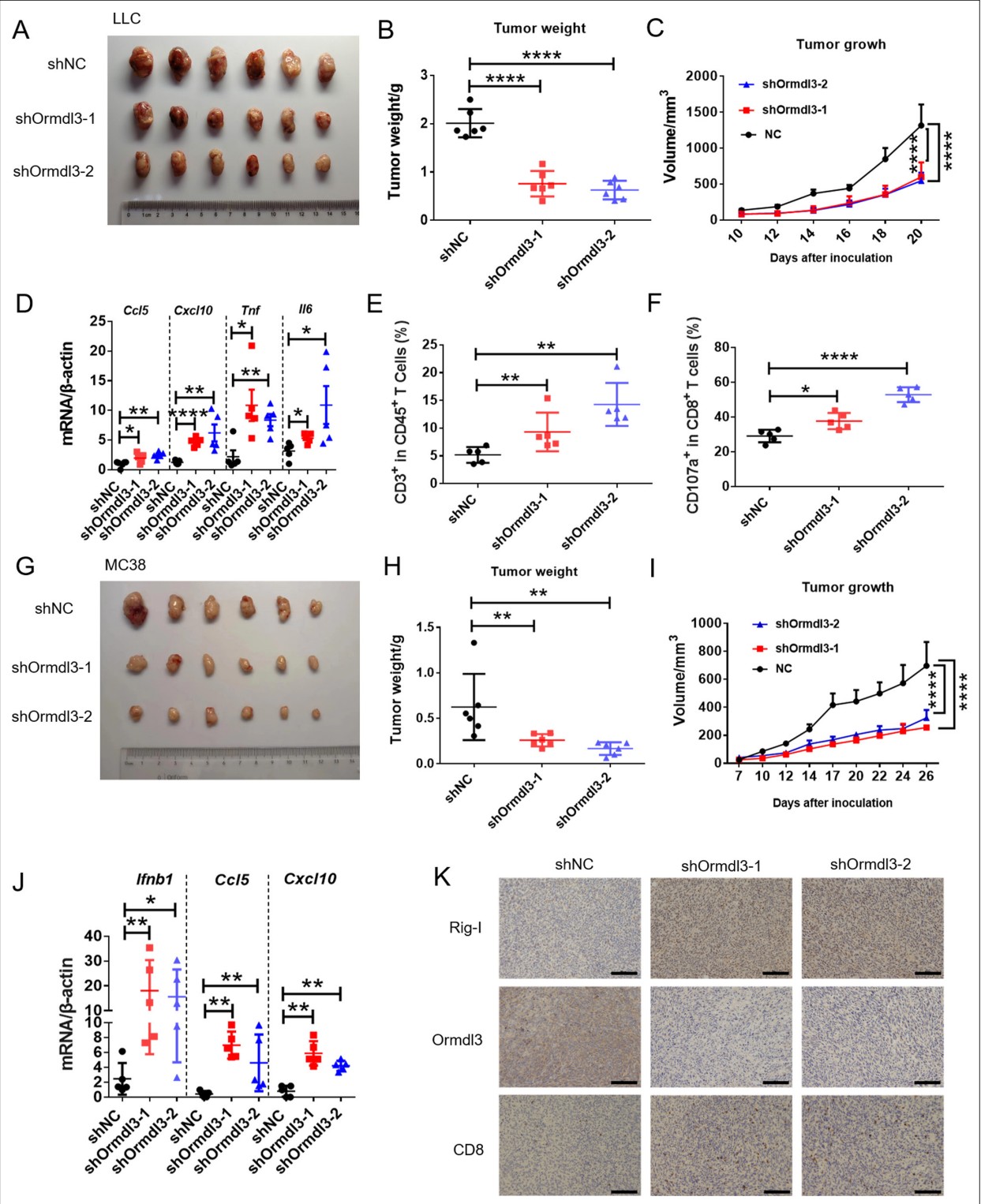

**Figure 7.** Knockdown of ORMDL3 in LLC/MC38 cells enhances anti-tumor immunity. (A–C) Representative images (A), tumor weight (B), and tumor growth (C) of LLC tumors on day 21 after inoculation with 1.5×10⁶ LLC cells with or without Ormdl3 stable knockdown into C57BL/6 mice, (n=6). (D) Results of the quantitative real-time PCR (qRT-PCR) assays showing mRNA levels of *Ccl5*, *Cxcl10*, *Tnf*, and *Il6* in LLC tumors, (n=5). (E, F) Flow cytometry assay of CD3⁺ T and CD107a⁺ CD8⁺ T cell percentages in indicated population, (n=5). (G–I) Representative images (G), tumor weight (H), and tumor growth (I) on day 27 after tumor inoculation with 5×10⁵ MC38 cells with or without Ormdl3 stable knockdown into C57BL/6 mice, (n=6). (J) Results of the qRT-PCR assays showing mRNA levels of *Ifnb1*, *Ccl5*, and *Cxcl10* in MC38 tumors, (n=5). (K) Results of the immunohistochemistry (IHC) assay showing

*Figure 7 continued on next page*

*Figure 7 continued*

expression levels of Ormdl3, Rig-I, and CD8 in MC38 tumors. Scale bars, 50 µm. Data from three independent experiments are presented as mean ± SD and were analyzed by one-way ANOVA (B, D-F, H, and J) or two-way ANOVA (C ,and I), *p<0.05, **p<0.01, ****p<0.0001.

The online version of this article includes the following source data and figure supplement(s) for figure 7:

**Figure supplement 1.** Inhibiting ORMDL3 increases the abundance of RIG-I.

**Figure supplement 1—source data 1.** Original files for western blots shown in *Figure 7—figure supplement 1*, indicating relevant bands.

**Figure supplement 1—source data 2.** Original files for western blots shown in *Figure 7—figure supplement 1*.

**Figure supplement 2.** ORMDL3 level is associated with poor survival and reduced immune cell infiltration and interferon-stimulated gene (ISG) expression.

**Figure supplement 3.** Schematic shows that ORMDL3 promotes the degradation of RIG-I and attenuates type I IFN in cancer.

as RNF55), and carboxyl terminus of HSC70-interacting protein (CHIP), mediate K48-linked ubiquitination of RIG-I, leading to its degradation through ubiquitin-proteasome pathway and attenuating cascade activation. Additionally, some deubiquitylating enzymes play significant roles in regulating RIG-I activity. For example, USP3, USP21, and CYLD remove K63-linked polyubiquitin chains, thereby repressing RIG-I activity. In contrast, USP4 and USP15 enhance the stability of RIG-I by reducing K48-linked ubiquitylation. Our study discovered that USP10 stabilizes RIG-I by decreasing its K48-linked ubiquitination at lysine residues 146, 154, 164, and 172.

Research on ORMDL3 has primarily focused on its relationship with asthma and rhinovirus infections. Mechanistic studies have revealed that ORMDL3 regulates intracellular adhesion molecule 1 expression (ICAM1) and modulates ceramide and sphingolipid metabolism (*Zhang, 2021*). However, the effects of ORMDL3 on human innate immunity remain to be elucidated further. In our study, we found that the transcription of *IFNB1* was significantly impaired in ORMDL3-overexpressing cells in response to viral infection (*Figure 1A*). Conversely, their transcription induced by RIG-I-N was significantly increased when ORMDL3 was knocked down (*Figure 2C*). These findings confirm that ORMDL3 is a negative regulator of antiviral innate immunity. ORMDL3 is a multiple transmembrane structural protein located at the core of the serine palmitoyltransferase (SPT) complex, where it stabilizes SPT assembly (*Li et al., 2021*). Co-IP experiments showed that the ER protein ORMDL3 interacts with mitochondrial protein MAVS, suggesting the existence of scaffold proteins mediating this interaction. ORMDL3 has also been implicated in calcium transport (*Carreras-Sureda et al., 2013*; *Cantero-Recasens et al., 2010*). Since calcium transfer between the ER and mitochondria plays an important role in protein synthesis, it is plausible that ER-mitochondria contact site proteins mediate the interaction between ORMDL3 and MAVS. Furthermore, we discovered that the deubiquitinating enzyme USP10 stabilizes RIG-I, while ORMDL3 disturbs this process, thereby negatively regulating type I IFN production. When USP10 was knocked down, ORMDL3 was unable to degrade RIG-I-N, indicating that USP10 is indispensable for ORMDL3-mediated RIG-I degradation.

Collectively, our findings identify ORMDL3 as a negative regulator of type I IFN pathway and antitumor immunity. The proposed working model is that ORMDL3 forms a complex with MAVS and promotes the degradation of RIG-I, thereby attenuating the transcription of type I IFN and cytotoxic CD8[+] T cell-mediated tumor killing (*Figure 7—figure supplement 3*). This negative regulatory loop of antiviral innate immunity mediated by ORMDL3 may provide insights for the development of therapeutics against viral infections and tumors.

## Materials and methods

**Key resources table**

| Reagent type (species) or resource | Designation | Source or reference | Identifiers | Additional information |
|---|---|---|---|---|
| Cell line (*Homo sapiens*) | HEK293T | ATCC | Cat#CRL-11268 | RRID:CVCL_1926 |
| Cell line (*Homo sapiens*) | A549 | ATCC | Cat#CCL-185 | RRID:CVCL_0023 |
| Cell line (*Homo sapiens*) | HeLa | ATCC | Cat#CCL-2 | RRID:CVCL_0030 |
| Cell line (*Homo sapiens*) | DLD1 | ATCC | Cat#CCL-221 | RRID:CVCL_0248 |

*Continued on next page*

*Continued*

| Reagent type (species) or resource | Designation | Source or reference | Identifiers | Additional information |
|---|---|---|---|---|
| Cell line (*Homo sapiens*) | HCT15 | ATCC | Cat#CCL-225 | RRID:CVCL_0292 |
| Cell line (*Homo sapiens*) | SW480 | ATCC | Cat#CCL-228 | RRID:CVCL_0546 |
| Cell line (*Homo sapiens*) | SW620 | ATCC | Cat#CCL-227 | RRID:CVCL_0547 |
| Cell line (*Homo sapiens*) | THP1 | This paper | TIB-202 | Cell line maintained in Shuai Chen lab (Sun Yat-sen University Cancer Center) |
| Cell line (*Mus musculus*) | MC38 | This paper | Gift from Prof. Penghui Zhou (Sun Yat-sen University Cancer Center) | Cell line maintained in Shuai Chen lab (Sun Yat-sen University Cancer Center) |
| Cell line (*Mus musculus*) | LLC | This paper | Gift from Prof. Penghui Zhou (Sun Yat-sen University Cancer Center) | Cell line maintained in Shuai Chen lab (Sun Yat-sen University Cancer Center) |
| Sequence-based reagent | *ORMDL3*-shRNA-1 | This paper | PCR primers | CCGGCCCACAGAATGTGATAGTAATCT CGAGATTACTATCACATTCTGTGGGTTTTTG |
| Sequence-based reagent | *ORMDL3*-shRNA-2 | This paper | PCR primers | CCGGCATGGGCATGTATATCTTCCTCTCGA GAGGAAGATATACATGCCCATGTTTTTG |
| Sequence-based reagent | *USP10* shRNA-1 | This paper | PCR primers | CCGGCCTATGTGGAAACTAAGTATTCTCGA GAATACTTAGTTTCCACATAGGTTTTTG |
| Sequence-based reagent | *USP10* shRNA-2 | This paper | PCR primers | CCGGCCCATGATAGACAGCTTTGTTCTC GAGAACAAAGCTGTCTATCATGGGTTTTTG |
| Sequence-based reagent | *USP10* shRNA-3 | This paper | PCR primers | CCGGCGACAAGCTCTTGGAGATAAACTC GAGTTTATCTCCAAGAGCTTGTCGTTTTTG |
| Sequence-based reagent | *Ormdl3*-shRNA-1 | This paper | PCR primers | CCGGCCAAGTATGACCAAGTCCATTCT CGAGAATGGACTTGGTCATACTTGGTTTTTG |
| Sequence-based reagent | *Ormdl3*-shRNA-2 | This paper | PCR primers | CCGGGCCGACTTGGAGTAGCTTGTACTCG AGTACAAGCTACTCCAAGTCGGCTTTTTG |
| Antibody | ORMDL3 (Rabbit, polyclonal) | Abcam | ab211522 | RRID:AB_3102000 WB (1:1000) |
| Antibody | ORMDL3 (Rabbit, polyclonal) | Abcam | ab107639 | RRID:AB_10863267 WB (1:1000) |
| Antibody | RIG-I (Mouse, monoclonal) | Santa Cruz | sc376845 | RRID:AB_2732794 WB (1:1000) |
| Antibody | Flag (Mouse, monoclonal) | Sigma | F1804# | RRID:AB_262044 WB (1:1000) |
| Antibody | GFP (Mouse, monoclonal) | Proteintech | 66002-1-Ig | RRID:AB_11182611 WB (1:1000) |
| Antibody | Myc (Mouse, monoclonal) | Proteintech | 60003-2-Ig | RRID:AB_2734122 WB (1:1000) |
| Antibody | Myc (Rabbit, polyclonal) | Proteintech | 10828-1-AP | RRID:AB_2148585 WB (1:1000) |
| Antibody | HA (Mouse, monoclonal) | Ray antibody | RM1004 | WB (1:1000) |
| Antibody | USP10 (Rabbit, polyclonal) | Abclonal | A13387 | RRID:AB_2760247 WB (1:1000) |
| Antibody | Tubulin (Mouse, polyclonal) | Fdbio | FD0064 | RRID:AB_3076327 WB (1:2000) |
| Antibody | GAPDH (Mouse, monoclonal) | Proteintech | 60004-1-Ig | RRID:AB_2107436 WB (1:2000) |

*Continued*

| Reagent type (species) or resource | Designation | Source or reference | Identifiers | Additional information |
|---|---|---|---|---|
| Antibody | GAPDH (Rabbit, monoclonal) | ServiceBio | GB15004 | RRID:AB_2943040<br>WB (1:5000) |
| Antibody | β-Actin (Mouse, monoclonal) | Proteintech | 60008-1-Ig | RRID:AB_2289225<br>WB (1:2000) |
| Antibody | β-Actin (Rabbit, monoclonal) | ServiceBio | GB15003 | RRID:AB_3083699<br>WB (1:2000) |
| Antibody | IRF3 (Rabbit, monoclonal) | Cell Signaling Technology | # 4302 | RRID:AB_1904036<br>WB (1:1000) |
| Antibody | Anti-phospho-IRF-3 (Rabbit, monoclonal) | Cell Signaling Technology | #29047 | RRID:AB_2773013<br>WB (1:1000) |
| Antibody | CD3ε-APC (anti-Mouse) | BioLegend | 100235 | RRID:AB_2561455<br>Flow (1:300) |
| Antibody | CD4-Pacific blue (anti-Mouse) | BioLegend | 100428 | RRID:AB_493647<br>Flow (1:300) |
| Antibody | CD8-PE-cy7 (anti-Mouse) | BioLegend | 100721 | RRID:AB_312760<br>Flow (1:300) |
| Antibody | CD45-APC-cy7 (anti-Mouse) | BioLegend | 157024 | RRID:AB_2876533<br>Flow (1:300) |
| Antibody | CD44-FITC (anti-Mouse) | BioLegend | 103006 | RRID:AB_312957<br>Flow (1:300) |
| Antibody | CD107a-PE (anti-Mouse) | BioLegend | 121611 | RRID:AB_17320511<br>Flow (1:300) |
| Chemical compound, drug | Poly(I:C) (LMW) | Invivogen | tlrl-picw | |
| Chemical compound, drug | Poly(dG:dC) | Invivogen | tlrl-pgcn | |
| Chemical compound, drug | SB9200 | Bidepharm | CAS:942123-43-5 | |
| Software, algorithm | GraphPad Prism 7 | GraphPad | N/A | |
| Software, algorithm | ImageJ | https://imagej.net/Fiji/Downloads | N/A | |
| Software, algorithm | FlowJo10 | FlowJo | N/A | |
| Commercial assay or kit | Lipo293 | Beyotime | C0521 | |
| Commercial assay or kit | Lipofectamine 2000 | Thermo Fisher Scientific | Cat#11668019 | |
| Commercial assay or kit | Lipofectamine RNAiMAX | Thermo Fisher Scientific | Cat#13778075 | |
| Commercial assay or kit | Dual-Luciferase Reporter Assay System | Promega | Cat#E1960 | |

## Cell lines

HEK293T cell lines (from embryonic kidney of female human fetus) were cultured at 37°C under 5% $CO_2$ in Dulbecco's modified Eagle's medium (DMEM) supplemented with 10% fetal bovine serum (FBS) (ExCell, FSP500), and A549 cell lines (from lung of a 58-year-old male human) were cultured in Roswell Park Memorial Institute (RPMI) 1640 medium. LLC and MC38, HCT15, DLD1, SW480, SW620 cell lines were obtained from American Type Culture Collection (ATCC) cultured in RPMI 1640 medium. The cell lines used in this study have been authenticated. Mycoplasma contamination was routinely checked by PCR analysis and eliminated by treatment with Plasmocin (ant-mpt). The primers

were as follows: Myco forward 5'- GGGAGCAAACAGGATTAGATACCCT-3'; Myco reverse 5'-GCAC CATCTGTCACTCTGTTAACCTC-3'.

## Viruses

VSV-GFP was provided by Prof. Rongfu Wang (Zhongshan School of Medicine, Sun Yat-sen University, China) and amplified in Vero cells. HSV-1-GFP was provided by Prof. Musheng Zeng (Sun Yat-sen University Cancer Center, China) and amplified in Vero cells. Cell lines were infected with VSV (MOI = 0.01), HSV-1 (MOI = 0.1) for various times, as indicated in the Figures.

## Plasmids and transfection

Expression plasmids for *RIG-I*, *MDA5*, *MAVS*, *TBK1*, *TRIF*, *IKBKE*, *IRF3*, and *IFN-β-luc*, *ISRE-luc* were provided by Prof. Xin Ye (Institute of Microbiology, Chinese Academy of Sciences). Plasmid encoding *ORMDL3* was cloned in pCMV-HA/ pCMV-Myc/ pCMV-Flag vector, and ORMDL3 truncations ORMDL3 (1–42aa), ORMDL3 (43–82aa), ORMDL3 (83–118aa), and ORMDL3 (119–153aa) were constructed into the pEGFP-N1 vector (Clontech Laboratories). Expression plasmids for *USP10* was obtained from Prof. Jing Tan (Sun Yat-sen University Cancer Center) and cloned into pCMV-Myc vector. For the transfection of plasmids, poly(I:C) (LMW) and poly(dG:dC) used in this study into HEK293T, A549, and BMDM cells, DNA Transfection Reagent PEI MW40000 (pH 7.1, Yeasen Biotechnology, 40816ES03), Lipo293 (Beyotime, C0521), Lipofectamine 2000 (Thermo Fisher Scientific), Lipofectamine 3000 (Thermo Fisher Scientific), or RNAiMAX (Thermo Fisher Scientific) were used according to the manufacturer's protocols.

## Flow cytometry

Single-cell suspensions were prepared from the tumor tissues of mice. Tumor tissues were cut into small pieces and washed with PBS containing 2% FBS. The tumors were digested in 15 ml RPMI supplemented with 2% FBS, 50 U/ml Collagenase Type IV (Invitrogen, CA, USA), 20 U/ml DNase (Roche, Indianapolis, IN) and incubated at 37°C for 30 min to 1 hr while gently shaking. Digested tumors were then filtered through a 70 µm strainer after washing three times with PBS. Red blood cells were lysed for 3–5 min with ACK lysis buffer and then washed with PBS containing 2% FBS. Single cells were stained with the appropriate antibodies to surface markers at 4°C for 30 min in the dark. The following fluorescent dye-labeled antibodies purchased from BD Biosciences, BioLegend, or Invitrogen were used in this study: CD3ε-APC (145-2C11), CD4-Pacific blue (GK1.5), CD8-PE-cy7 (KT15), CD45-APC-cy7 (30-F11), CD44-FITC (IM7), CD107a-PE (1D4B). All flow cytometric data were collected on BD Fortessa X20 (BD Biosciences, San Jose, CA, USA) and performed using FlowJo analysis software v10.4. LLC tumor tissues were grinded into a single-cell suspension and treated as described above.

## Tumor models

$1.5×10^6$ LLC cells were subcutaneously implanted into the flanks of C57BL/6 mice. $5×10^5$ MC38 cells were implanted the same as LLC cells; after the tumor was established, the volume of tumor was measured once every 2 days.

## Mass spectrometry and co-IP

$1×10^7$ HEK293T cells transfected with flag-vector or flag-ORMDL3 were prepared by washing with cold PBS and then lysed with 1× lysis buffer (Cell Signaling Technology) and incubated on ice for 30 min. Supernatants were collected and immunoprecipitated with the indicated antibodies for 4 hr at 4°C, recovered by adding protein A/G Sepharose Beads (Santa Cruz Biotechnology, CA, USA, #sc-2003) overnight. After incubation, beads were washed with wash buffer and immersed in PBS then subjected to mass spectrometry. Immunoblot assays were performed with specific antibodies to identify the proteins interacting with ORMDL3. The following antibodies were used for co-IP or immunoblot assay: ORMDL3 (Abcam, 107639) (Abcam, 211522), Flag (Sigma, St. Louis, MO, USA, #F1804), HA (Beijing Ray Antibody Biotech, RM1004). The mouse antibody GAPDH (1:2000 for immunoblot, 60004-1-Ig) and rabbit antibody beta-Actin (1:2000 for immunoblot, #GB15003) were purchased from Servicebio Biotechnology (Wuhan, China). Rabbit anti-GFP (50430-2-AP), mouse anti-GFP (66002-1-Ig), anti-β-tubulin (66009-1-Ig), rabbit anti-Myc (10828-1-AP), mouse anti-Myc (60003-2-Ig) were ordered from

Proteintech. Mouse anti-RIG-I (sc376845) were ordered from Santa Cruz, rabbit anti-USP10 (A13387) were purchased from Abclonal. Secondary antibodies were purchased from The Jackson Laboratory. The 10× cell lysis buffer (#9803) was bought from Cell Signaling Technology. Protein Marker (DB180-10) was bought from MIKX. The protease inhibitor cocktail and phosphatase inhibitor cocktail were purchased from TargetMol (C0001).

## Dual-luciferase reporter assay

HEK293T cells were transfected with plasmids encoding IFN-β or ISRE luciferase reporter and RIG-I(N), MDA5, MAVS, TBK1, IRF3-5D together with pRL-TK and the plasmid encoding ORMDL3. Cells were collected and lysed 24 hr post-transfection. Subsequently, the luciferase activities were measured using a Dual-Luciferase Reporter Assay System (Promega, Madison, WI, USA, #E1910). Data were normalized by the ratio of Firefly luciferase activity to Renilla luciferase activity. Each group was measured in triplicate.

## RNAi

All the siRNA oligonucleotides containing 3′dTdT overhanging sequences were chemically synthesized in GenePharma (Suzhou, China) and transfected into cells using Lipofectamine RNAiMAX Transfection Reagent (Thermo Fisher). The siRNAs corresponding to the target sequences were synthesized in RIBOBIO (Guangzhou, China). In this study, the siRNAs' sequences were designed as follows: *ORMDL3* si #1, 5′-GCAUCUGGCUCUCCUACGUUTdTdT 3′; *ORMDL3* si #2, 5′- GGCAAGGCGAGGC UGCUAAUTdTdT 3′; *ORMDL3* si #3, 5′- CCCUGAUGAGCGUGCUUAUTdTdT 3′. For transfection of siRNAs used in this study into HEK293T cells, Transfection Reagent Lipofectamine RNAiMAX (Thermo Fisher Scientific) was used according to the manufacturer's protocol.

## RNA extraction and qRT-PCR

Total RNA from cells was extracted with TRIzol reagent (Vazyme R711) according to the manufacturer's instructions. Tumor tissue RNA was grinded by a sample freezing grinder: LUCA (LUKYM-I). Complementary DNA was synthesized using the HiScript II Q RT SuperMix (Vazyme, R223-01). SYBR Green Mix (GenStar, A301-10) was used for qRT-PCR assays. Relative quantification was performed with the $2^{\wedge}(-\Delta\Delta CT)$ method using 18S or β-Actin for normalization. The specific qRT-PCR primers are listed in *Supplementary file 1, table 1*.

## Establishment of overexpressed stable cell lines and knockdown cell lines

ORMDL3 cDNA was constructed into the pCDH-CMV-MCS-EF1 vector. A549 and HEK293T cells which stably overexpress plasmid encoding ORMDL3 were generated by lentivirus-mediated gene transfer. HEK293T cells were co-transfected with lentiviral expressing plasmid, lentiviral packaging plasmid psPAX2 (Addgene, #12260), and VSV-G envelope expressing plasmid pMD2.G (Addgene, #12259). After 48 hr, the lentiviruses were used for infecting A549 cells and then selected cells with puromycin (Thermo Fisher Scientific). Human ORMDL3 knockdown cell lines: shORMDL3-1 and shORMDL3-2 stable cell lines. Human USP10 knockdown cell lines: shUSP10-1 and shUSP10-2, and shUSP10-3 stable cell lines. Murine Ormdl3 knockdown cell lines: shOrmdl3-1 and shOrmdl3-2 stable cell lines. Annealing oligos were ligated into PLKO.1 vector. After lentivirus packaging, the target cells were then selected with puromycin. The shRNA sequences are listed as follows.

> sh*ORMDL3*-1:   CCGGCCCACAGAATGTGATAGTAATCTCGAGATTACTATCACATTCTGTGGG TTTTTG;
> sh*ORMDL3*-2:   CCGGCATGGGCATGTATATCTTCCTCTCGAGAGGAAGATATACATGCCCATG TTTTTG;
> sh*USP10*-1: CCGGCCTATGTGGAAACTAAGTATTCTCGAGAATACTTAGTTTCCACATAGGTTTT TG;
> sh*USP10*-2:   CCGGCCCATGATAGACAGCTTTGTTCTCGAGAACAAAGCTGTCTATCATGGG TTTTTG;
> sh*USP10*-3:   CCGGCGACAAGCTCTTGGAGATAAACTCGAGTTTATCTCCAAGAGCTTGTCG TTTTTG;

sh*Ormdl3*-1:    CCGGCCAAGTATGACCAAGTCCATTCTCGAGAATGGACTTGGTCATACTTGG
TTTTTG;

shary*Ormdl3*-2:    CCGGGCCGACTTGGAGTAGCTTGTACTCGAGTACAAGCTACTCCAAGTCGGC
TTTTTG.

## BMDM

Macrophages were differentiated from the bone marrow of wild-type (WT) C57BL/6 mice. All bone marrow cells were flushed out and filtered through a 70 µm cell strainer. After centrifugation, red blood cells were lysed. The resultant bone marrow cells were resuspended in RPMI 1640 (Gibco) supplemented with 10% FBS (Gibco), 1% penicillin/streptomycin (Gibco), and 50 µM 2-mercaptoethanol (Sigma) in the presence of 20 ng/ml macrophage colony-stimulating factor (M-CSF, PeproTech) for 7 days and mature BMDM were stimulated with indicated stimulation of poly(I:C) or poly(dG:dC).

## AAV virus production

All AAV vectors were produced in HEK293T cells via the triple plasmid transient transfection methods as pAdDelta F6 (Addgene, #112867): pAAV2/1 (Addgene, #112862): target gene = 2:1.2:1. For small-scale preps, HEK293T cells were seeded in 10 cm dishes and grown to 80% confluence in DMEM containing 10% FBS (Gibco, 26140079) and 1% PenStrep (Thermo Fisher Scientific, 15140122). Cells were then triple transfected with the Ad helper plasmid pAdDelta F6, AAV2/1 Rep/Cap, pscAAV-CAG-GFP (Addgene, #83279), or pscAAV-CAG-Ormdl3, at a ratio of 2:1.2:1 (7.5:4.5:3.75 µg per 10 cm dish) using PEI MW40000 at a ratio 4:1 of PEI/total DNA. Cells were harvested 3 days post-transfection by scraping cells off the plate in their conditioned medium and lysing cells through 3× freeze-thaw cycles between 37°C and –196°C. Preps from three replicate plates were then pooled and incubated with 25 U/mL of UCF.ME Nuclease Ultrapure (Yeasen Biotechnology, 20125ES24-25KU) at 37°C for 1 hr to remove plasmid and cellular DNA, centrifuged at 4°C and 14,000×$g$ for 30 min, and the supernatant filtered through a 0.22 µm polyethersulfone bottle-top filter (Corning, 431097). The filtered lysate was purificated by iodixanol gradient ultracentrifugation. For AAV collection, the fractions obtained from the 40% phase were analyzed by measuring absorbance at 20-fold dilution at 340 nm to identify the main contaminating protein peak, as previously described. For ultrafiltration/concentrated AAV, 0.001% Pluronic F68+200 mM NaCl PBS was added to the pool to reach a total volume of 15 ml, using Amicon Ultra-15 centrifugal filter units (MWCO, 100 kDa; Merck Millipore). After concentration to a minimum of 500 µl, the product was aliquoted and stored at –80°C.

## AAV titration

Prepare a plasmid stock of 2×10$^9$ molecules/µl to generate a standard curve, and then treat the purified AAV samples with DNase I to eliminate any contaminating plasmid DNA carried over from the production process (DNase does not penetrate the virion). Make six serial dilutions of the reference sample, DNase-treated and AAV samples and detect them with qRT-PCR. Then, perform data analysis using the instrument's software. Determine the physical titer of the samples (viral genomes, vg/ml) based on the standard curve and the sample dilutions.

## Immunofluorescence labeling and confocal microscopy for FRET assay

YFP-MAVS and CFP-ORMDL3 expressing plasmids were co-transfected into HeLa cells and incubated for 24 hr. Cells were fixed with 4% paraformaldehyde for 30 min. Images were observed on laser confocal fluorescence microscopy (Zeiss, LSM880), and we bleached YFP-MAVS and measured the fluorescence intensity of YFP-MAVS and CFP-ORMDL3.

## Co-IP and immunoblot analysis

For IP, cells were lysed with lysis buffer (Cell Signaling Technology) supplemented with protease inhibitor for 30 min at 4°C. After centrifugation at 12,000 rpm and 4°C for 10 min, supernatants were collected and incubated with appropriate antibodies for 1 hr and protein A/G beads (Santa Cruz Biotechnology) overnight. Thereafter, the beads were washed four times with cold PBS, followed by SDS-PAGE and immunoblot analysis. For immunoblot analysis, cells or tissues were lysed with RIPA buffer (Cell Signaling Technology). Protein concentrations were measured with Bradford Protein Assay Kit (Beyotime), and equal amounts of lysates were used for SDS-PAGE. The samples were

eluted with SDS loading buffer by boiling for 10 min and then performed SDS-PAGE. The proteins were transferred onto PVDF membrane (Roche), and immunoblot analysis was performed with appropriated primary antibodies at 4°C overnight and horseradish peroxidase-conjugated secondary anti-mouse or anti-rabbit antibodies for 1 hr at room temperature. ChemiDoc Touch (Bio-Rad) achieved visualization.

## Statistical analysis

The data were analyzed with GraphPad Prism 7. For two independent groups, the Student's t test was used to determine statistical significance. Statistical details for individual experiments can be found in the Figure legends. Statistical significance was two-tailed and $p < 0.05$ is considered statistically significant. p-Values are indicated by asterisks in the figures as follows: $*p < 0.05$, $**p < 0.01$, $***p < 0.001$, $****p < 0.0001$, and ns indicates no significance.

## Acknowledgements

This project was supported by grants from the National Natural Science Foundation of China (82273045), the Guangdong Basic and Applied Basic Research Foundation (2022A1515011930), and the Guangzhou Basic and Applied Basic Research Foundation (2023A04J2118).

---

## Additional information

### Funding

| Funder | Grant reference number | Author |
|---|---|---|
| National Natural Science Foundation of China | 82273045 | Shuai Chen |
| Guangdong Basic and Applied Basic Research Foundation | 2022A1515011930 | Shuai Chen |
| Guangzhou Basic and Applied Basic Research Foundation | 2023A04J2118 | Chunjie Sheng |

The funders had no role in study design, data collection and interpretation, or the decision to submit the work for publication.

### Author contributions

Qi Zeng, Data curation, Methodology, Writing – original draft, Project administration, Writing – review and editing; Chen Yao, Shimeng Zhang, Yizhi Mao, Jing Wang, Ziyang Wang, Methodology; Chunjie Sheng, Methodology, Writing – review and editing; Shuai Chen, Supervision, Funding acquisition, Writing – review and editing

### Author ORCIDs

Qi Zeng https://orcid.org/0009-0001-9722-5526
Chunjie Sheng https://orcid.org/0000-0002-4715-7617
Shuai Chen https://orcid.org/0000-0003-1280-018X

### Ethics

All animal studies in this study were carried out according to the National Institute of Health Guide for the Care and Use of Laboratory Animals with the approval of Sun Yat-Sen University Cancer Center Institutional Animal Care and Use Committee (approval number: SYSU-IACUC-2023-000104).

Reviewer #2 (Public review): https://doi.org/10.7554/eLife.101973.3.sa1
Author response https://doi.org/10.7554/eLife.101973.3.sa2

---

# Additional files

## Supplementary files
Supplementary file 1. Primers for qPCR. Related to Materials and methods.

MDAR checklist

## Data availability
Raw data that support the findings of this study has been deposited in figshare at https://doi.org/10.6084/m9.figshare.28646573. All data generated or analysed during this study are included in the manuscript and supporting files; source data files have been provided for Figures 1–6 and the figure supplements.

The following dataset was generated:

| Author(s) | Year | Dataset title | Dataset URL | Database and Identifier |
|---|---|---|---|---|
| Zeng Q, Yao C, Zhang S, Mao Y, Mao Y, Wang J, Wang Z, Sheng C, Chen S | 2025 | Datasets for the eLife manuscript "ORMDL3 restrains type I interferon signaling and anti-tumor immunity by promoting RIG-I degradation | https://doi.org/10.6084/m9.figshare.28646573 | figshare, 10.6084/m9.figshare.28646573 |

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
