## [Editor Report · eLife Assessment]

This study provides **important** insights into the regulation of type-I interferon signaling and anti-tumor immunity, demonstrating that ORMDL3 promotes RIG-I degradation to suppress immune responses. The evidence is **convincing**, with well-executed mechanistic experiments and in vivo validation in syngeneic tumor models. These findings have significant implications for cancer immunotherapy, highlighting ORMDL3 as a potential therapeutic target.

---

## [Referee Report · Reviewer #2 (Public review)]

Summary:

The authors identified ORMDL3 as a negative regulator of the RLR pathway and anti-tumor immunity. Mechanistically, ORMDL3 interacts with MAVS and further promotes RIG-I for proteasome degradation. In addition, the deubiquitinating enzyme USP10 stabilizes RIG-I and ORMDL3 disturbs this process. Moreover, in subcutaneous syngeneic tumor models in C57BL/6 mice, they showed that inhibition of ORMDL3 enhances anti-tumor efficacy by augmenting the proportion of cytotoxic CD8-positive T cells and IFN production in the tumor microenvironment (TME).

Strengths:

The paper has a clearly arranged structure and the English is easy to understand. It is well written. The results clearly support the conclusion.

Comments on revisions:

All questions have been answered.

---

## [Author Response]

The following is the authors’ response to the original reviews.

**Reviewer #1 (Recommendations for the authors):**
Minor Points:• HEK293T cells are not typically Type 1 IFN-producing cells; it is recommended to use other immune cell lines to validate results obtained with ORMDL3 overexpression in 293T cells. The same applies to A549 alveolar basal epithelial cells.

Thanks for the reviewer’s insightful comment. In Figure 1C, we overexpressed ORMDL3 in mouse primary BMDM cell and stimulated it with poly(I:C) or poly(dG:dC), which suggests that ORMDL3 inhibits IFN expression in primary cell BMDM.

• Clarify whether TLR3 is expressed in the cell lines used in Figure 1 and whether TLR3 is present in mouse BMDMs.

Thanks for your suggestions. We identified whether TLR3 is expressed in HEK293T, A549 and BMDM. We designed primers of human TLR3 and murine Tlr3, and the results showed that Tlr3 is expressed in BMDM but not in HEK293T and A549. As it shown in Author response image 1.

**Author response image 1. sa2fig1:** PCR amplification of human TLR3 was conducted on cDNA derived from HEK293T and A549 cells (lanes 1 and 2, respectively), and PCR amplification of murine Tlr3 was performed on cDNA from BMDM (lane 3). Human spleen cDNA (lane 4, TAKARA Human MTCTM Panel I, Cat# 636742) served as a positive control, and 18s rRNA was used as an internal control.

primer sequences:

human TLR3: forward TTGCCTTGTATCTACTTTTGGGG reverse TCAACACTGTTATGTTTGTGGGT

murine Tlr3: forward GTGAGATACAACGTAGCTGACTG reverse TCCTGCATCCAAGATAGCAAGT

18s (human/mice): forward GTAACCCGTTGAACCCCATT reverse CCATCCAATCGGTAGTAGCG

• Specify the type of luciferase reporter assay used in Figure 1E.

Thanks for the reviewer’s insightful comment. The Dual-Luciferase Reporter (DLR) Assay System efficiently measures two luciferase signals. In brief, the IFN-reporter luciferase is derived from firefly (Photinus pyralis), while the internal control luciferase is from Renilla (Renilla reniformis or sea pansy). These dual luciferases are measured sequentially from a single sample. In Figure 1E, we measured the luciferase activity of IFN (firefly) and internal control gene TK (Renilla), and their ratio is shown in Figure 1E.

• Clarify what was knocked down in the A549 stable KD cell line and whether HSV-1 infects and replicates in A549 cells.

We sincerely appreciate the reviewer’s concern and apologize for any ambiguous descriptions. In Figure 1H, we knocked down ORMDL3 and infected the cell with HSV-1, which shows that ORMDL3 does not affect the infection and replication of HSV-1 in A549.

• In Figure 2E, provide the rationale for using the same tag (Flag) in overexpression experiments with different molecules such as Flag-ORDML3 and Flag-RIG-I.

We thank the reviewer’s concern. We tried to co-express different tags of ORMDL3 and innate immunity proteins, and we got the same results as before. ORMDL3-Myc overexpression can only promote the degradation of Flag-RIG-I-N, as shown in current Figure 2E.

• Address the low knockdown efficiency shown in Figure 2D and consider whether it is sufficient for drawing conclusions.

Thanks for the reviewer’s concern. Because ORMDL3 antibody (Abcam 107639) can recognize all ORMDL family members (ORMDL1, 2 and 3), this may explain why the knockdown efficiency of ORMDL3 is not apparent in Figure2D. We also detect the knockdown efficiency of ORMDL3 at mRNA level, which showed that ORMDL3 was silenced efficiently and specifically (Figure S2C).

• Replace the Tubulin/β-Actin WB control with a more distinguishable band.

Thanks for the suggestion. Owing to different gel concentration, sometimes the protein bands appear fused, but it is distinguishable that the internal controls are consistent.

• In Figures 3D/E, the expression level of the Lysine mutant of RIG-I-N is too low. Please provide an explanation or repeat the experiment to achieve comparable expression levels and update the figure accordingly.

Thanks for the question. The expression of lysine mutant of RIG-I-N is low, we have increased the amount of plasmid in transfection, but this still hasn't increased its expression level. Though its abundance is low, we provided evidence to show that it would not be degraded by ORMDL3. In some literatures (for example: RNF122 suppresses antiviral type I interferon production by targeting RIG-I CARDs to mediate RIG-I degradation. Proc Natl Acad Sci U S A. 2016 Aug 23;113(34):9581-6**;** TRIM4 modulates type I interferon induction and cellular antiviral response by targeting RIG-I for K63-linked ubiquitination. J Mol Cell Biol. 2014 Apr;6(2):154-63.), it has also been reported that lysine mutant can affect RIG-I stability. In addition, we speculate that the 4KR mutant (K146R, K154R, K164R, K172R) may change RIG-I conformation, so its expression is lower.

• Explain why there is no difference in MAVS expression levels despite binding with MAVS.

Thanks for the question. In our experiment, ORMDL3 has no effect on MAVS expression. Our results showed that ORMDL3 interacts with MAVS and promotes the degradation of RIG-I, so only RIG-I level has a significant difference.

• Verify if Flag-tagged ORMDL3 is present in the IP sample in Figure 3G.

Thanks for the comment. We reloaded the samples and blot flag, and we found that ORMDL3 cannot be pulled down by RIG-I. We have added the results in Figure 3G.

• Reload the samples in Figure 4C to clearly identify the correct band for GFP-tagged ORMDL3.

Thanks for the question. As ORMDL3 is small molecular protein, we fused it and its fragments to GFP to increase its molecular weight. In our GFP vector, for some unknown reason, the 26kDa band always exists. This is actually a technical difficulty. Although the GFP-fused protein and GFP band are very close, they can still be distinguished as two bands.

• Rerun the Western blot for Actin IB in Figure 4E, as the ORMDL3-GFP (1-153) full-length appears abnormal.

Thanks for the question. As we first blot GFP and then blot actin on the same membrane, so it appears abnormal. We reloaded the previous sample and blotted the actin again.

• Clarify in which figure RIG-I ubiquitination is shown and whether ORMDL3 has E3 ubiquitin ligase activity. Explain how ORMDL3 facilitates USP10 transfer to RIG-I despite no direct interaction.

Thank you for your question. In Figure 3B we showed the ubiquitination of RIG-I and ORMDL3 does not have an E3 ubiquitin ligase activity. Our results showed that although ORMDL3 does not directly interacted with RIG-I, it forms complex with USP10 (Figure 5B, 5C) and disrupt USP10 induced RIG-I stabilization by decreasing the interaction between USP10 and RIG-I (Figure 6A). The detailed mechanism needs further investigation.

• Provide quantification for Figure 5D. Explain why the bands are not degraded by RIG-I and USP10.

Thanks for the concern. We quantified the bands and found that overexpression of USP10 increased RIG-I protein abundance. The quantitative gray values are added into the image. USP10 functions to stabilize RIG-I rather than promoting its degradation.

• Explain the decrease in RIG-I levels in Figure 5E when USP10 levels decrease.

Thanks for the concern. As shown in the working model (Supplementary Figure 8), USP10 is a deubiquitinase that stabilizes RIG-I by decreasing its K48-linked ubiquitination. So, in Figure 5E, we knocked down USP10 and found a decrease in RIG-I levels, which is consistent with Figure 5D.

• Clarify whether K48 ubiquitination on RIG-I has decreased in Figure 5F, as this is not clear from the image.

Thanks for the question. In Figure 5F it is shown that the K48 ubiquitination level of RIG-I significantly decreased (please see the density of the bands in the IP samples).

• Address whether ORMDL3 reduces RIG-I-N degradation in Figure 5H, as the results do not clearly support this claim.

Thanks for the concern. We quantified the bands and the results showed that ORMDL3 promotes the degradation of RIG-I-N. The quantitative gray values are added into the image.

• Reload Flag-ORMDL3 in Figure 6C to determine whether RIG-I-N is restored in the MG132-treated samples.

Thank you for your question. We quantified the bands and the results showed that RIG-I-N is restored in the MG132-treated samples. The quantitative gray values are added into the image.

• Correct numerous typos and errors, especially in the Discussion section, to improve readability

Thanks for the suggestion. We have revised the manuscript carefully to correct these errors.

**Reviewer #2 (Recommendations for the authors):**
(1) In Figure 1G and H, The number of virus-infected cells was observed using a fluorescence microscope. In addition, can the author use other techniques to detect the impact of ORMDL3 on virus replication?

Thanks for the question. Except for using a fluorescence microscope, we also used RT-PCR to quantify the amount of viral mRNA, and results were added in Figure 1G and H.

(2) In Figure 3C, ORMDL3 overexpression promotes the degradation of RIG-I-N. ORMDL3 is one of three ORMDL proteins with similar amino acid sequences, does ORMDL1/2 also have this function?

Thanks for the suggestion. We compared the function between ORMDLs and found that only ORMDL3 overexpression facilitated RIG-I-N degradation. The results were shown in Figure S2D.

(3) In Figure 5A, USP10 is not the top protein in the Mass spec assay. Does the author verified the interaction between ORMDL3 and other protein (for example CAND1)?

Thanks for your suggestion. We verified that ORMDL3 has no interaction with CAND1 and UFL1 but only interacts with USP10, as Figure S5 shows.

(4) A scale bar to be added to the images in Figure 1 G, H and Figure 7K.

Thanks for the suggestion. We have added the scale bars.

(5) The annotations in Figure 4B, C and E should be aligned.

Thanks for the suggestion. We have aligned the annotations.

(6) Provide Statistical methods

Thanks for the suggestion. We have provided the statistical methods in the materials and methods part.